# Biological Control and Cross Infections of the *Neofusicoccum* spp. Causing Mango Postharvest Rots in Spain

**Lucía Guirado-Manzano** [1,2], **Sandra Tienda** [1,2], **José Antonio Gutiérrez-Barranquero** [1,2], **Antonio de Vicente** [1,2], **Francisco M. Cazorla** [1,2] **and Eva Arrebola** [1,2,*]

1 Departamento de Microbiología, Faculta de Ciencias, Universidad de Málaga, Bulevar Louis Pasteur 31, 29010 Malaga, Spain; luguirado@uma.es (L.G.-M.); sandratienda@uma.es (S.T.); jagutierrez@uma.es (J.A.G.-B.); adevicente@uma.es (A.d.V.); cazorla@uma.es (F.M.C.)

2 Instituto de Hortofruticultura Subtropical y Mediterránea "La Mayora" IHSM-UMA-CSIC, 29010 Malaga, Spain

* Correspondence: ead@uma.es

**Abstract:** Mango is one of the main subtropical crops growing in southern Spain. Spanish mango fruit production can be efficiently transported to the rest of Europe, and these mangoes are very appreciated for their quality and flavour. However, postharvest rots have been detected in stored mango fruits, making their commercialization difficult. The causal agents associated with such rot symptoms have been isolated and identified. Because the mango crops used to share the same growing area with avocado crops, fungal presence on surrounding asymptomatic avocado fruits was also analysed to detect potential cross infections. Artificial inoculation with *Neofusicoccum parvum* and *N. mediterraneum* was able to reproduce rot symptoms in mango but was also able to induce rots in avocado fruits. To approach a biological control strategy against these rot-producing fungi, two very well-known antagonistic biocontrol bacteria, *Pseudomonas chlororaphis* PCL1606, and *Bacillus velezensis* UMAF6639, were tested. The obtained results revealed that both bacteria can control the fungal rots on stored mango and avocado fruits under controlled conditions. Additionally, the strain *B. velezensis* UMAF6639 showed the ability to persist on the fruit surface of adult commercial trees in experiments under open field conditions, helping to prevent the appearance of these postharvest diseases.

**Keywords:** postharvest disease; mango; avocado; *Pseudomonas chlororaphis*; *Bacillus velezensis*; *Neofusicoccum* spp.

## 1. Introduction

Mango (*Mangifera indica* L.) and avocado (*Persea americana* Mill.) crops are tropical crops that currently share their growth area in southern Spain. Mango crops currently reach approximately 5500 Ha, while avocado crops expand to approximately 15,000 ha [1]. Both crops are well adapted to growing in the coastal area of southern Spain because this geographical area has a subtropical microclimate that includes more than 300 days of sunshine per year and average annual temperatures of 17–20 °C (Meteorology Statal Agency, AEMET). Thus, these environmental conditions allow both crops to be cultivated in the same growing area, even sharing locality and farmlands, as well as some management processes of the fruit at the growers' associations.

This situation could lead to the appearance of different pathogens that could potentially affect these crops. The case of postharvest rots produced by fungi is relevant, especially those that could simultaneously affect both crops, being able to develop cross-infection events. In southern Spain, postharvest diseases can eventually be observed in mango fruits, but no postharvest diseases have been reported for avocados. Differences in processing and transport are minimal since both fruits are immediately processed and transported, minimizing the storage period and the appearance of postharvest diseases.

Thus, the presence of postharvest rots in mangoes could be related to the differences in physiological characteristics of the pulp and the skin of both fruits [2,3].

The main postharvest fungal disease that could affect mango and avocado fruits is anthracnose produced by *Colletotrichum* spp. [4]. This disease has been mainly attributed to *Colletotrichum gloeosporioides*, but other species of *Colletotrichum* can develop fruit rot disease, such as *C. asianum*, *C. fructicola*, *C. tropicale*, *C. dianesei*, and *C. karstii* [4,5]. Another important postharvest disease is stem-end rot [6], the causing agents of which include fungal species belonging to the Botryosphaeriaceae family, such as *Lasiodiplodia theobromae*, *Phomopsis mangiferae*, *Dothiorella dominicana* and *D. mangiferae*, *Cystophaera mangiferae*, *Neofusicoccum* spp., and *Pestalotiopsis* spp. Additionally, *Alternaria alternata* and *Colletotrichum gloeosporioides* can also be involved in stem-end rot symptoms [6–9].

Traditionally, chemical treatments have been used to combat fungal diseases; however, the regularization of the use of phytosanitary products as well as the eventual presence of residues on fruits and vegetables intended for human consumption have reduced the methods to mitigate losses produced by postharvest decay [10]. The current postharvest approaches include heat treatment, edible coating, irradiation, nitric oxide, sulfur dioxide, ozone, ethylene, 1-methylcyclopropene (1-MCP), controlled storage atmosphere and modified packaging atmosphere, calcium or salicylic acid application, storage in blue light, essential oils and plant extracts, inorganic salts such as potassium phosphite or sodium bicarbonate, and biological control [11,12].

Biological control by using microorganisms is being improved as an additional method to help prevent pre- and postharvest diseases [13]. Postharvest disease suppression mediated by antagonistic microorganisms is possible by employing naturally occurring epiphytic antagonistic microbiota from fruit and leaf surfaces [14,15]. Interestingly, antifungal compounds produced by biocontrol *Bacillus* spp. (such as lipopeptides) and *Pseudomonas* spp. (such as pyrrolnitrin) have been active against important phytopathogenic fungi, including those that cause postharvest diseases [16,17].

Antagonistic *Pseudomonas chlororaphis* (Pc) strain PCL1606 and *Bacillus velezensis* (Bv) strain UMAF6639 are two bacteria isolated from the same region where mango and avocado crops are both cultivated in southern of Spain. The Pc strain was isolated from the roots of healthy avocado trees growing in a *Rosellinia necatrix* (a causal agent of white root rot in avocado)-infested area. Pc produces the antimicrobial compounds 2-hexyl, 5-propyl resorcinol (HPR), pyrrolnitrin and hydrocyanic acid but does not produce phenazines [18,19]. Bv was isolated from the surface of a greenhouse-grown melon plant leaf. Its antagonist ability is mediated mainly by antagonistic lipopeptide production, which allows Bv to be an effective inhibitor of powdery mildew [20] and other postharvest fungi such as *Alternaria citri*, *Colletotrichum gloeosporioides*, or *Lasiodiplodia theobromae* [21].

The main goal of the current study is to detect and describe the main causal agents of the mango rots in southern Spain, to test whether they can also be detected on avocado fruits and whether they can cause cross infections and rot symptoms on mango and avocado fruits, and, finally, to approach a biological control strategy to prevent these postharvest pathogenic fungi by using the biocontrol agents Pc and Bv under controlled and open field conditions.

## 2. Materials and Methods

### 2.1. Fungal and Bacterial Isolates Used in This Study

A fungal collection associated with mango fruits displaying rot symptoms was constructed. Additionally, fungal isolations from asymptomatic avocado fruits growing in the same geographical area were also obtained. Mango fruits with visible rot symptoms were received from October 2019 to January 2020 (Figure S1, Supplemental Materials) and processed as follows. The whole fruits were first washed in tap water to remove the dust, and then superficially disinfected (10:20:70 of alcohol-commercial bleach-distilled water) for 3 min, followed by two consecutive rinses (of 3 min each) in sterile distilled water. Then, the surface-disinfected mango fruits were dried up and small fragments (1 cm$^2$),

including healthy and rotten areas, were trimmed. Those pieces of mango were deposited in acidified potato dextrose agar (PDA) plates (Oxoid, Thermo Fisher Scientific, Swindon, UK) supplemented with 1 mL of 85% lactic acid, to avoid bacterial development [22]. After seven days of incubation at 25 °C, the fungal mycelia growing from the fruit samples were taken by obtaining a portion of a single hypha using a binocular loupe and isolated in pure culture for further identification. This fungal collection also included ten fungal isolates previously obtained in 2013 from mango fruits with similar rot symptoms (Table 1).

**Table 1.** List of fungal isolates obtained from mango and avocado skin. Strains codes and ITS accession numbers of the isolates used in this study.

| Fungal Isolates Code and Accession Number | | Source Fruit | Date of Isolation | Reference |
|---|---|---|---|---|
| Genus *Alternaria* | | | | |
| UMAF M1306 | MZ160921 | Mango | February 2013 | This study |
| UMAF M1310 * | MZ160925 | Mango | February 2013 | This study |
| UMAF M1913 * | MZ160928 | Mango | October 2019 | This study |
| UMAF M1915 | MZ160930 | Mango | October 2019 | This study |
| UMAF M1916 | MZ160931 | Mango | October 2019 | This study |
| UMAF M1918 | MZ160933 | Mango | October 2019 | This study |
| UMAF M1931 * | MZ160946 | Mango | December 2019 | This study |
| UMAF M1932 | MZ160947 | Mango | December 2019 | This study |
| UMAF M1933 * | MZ160948 | Mango | December 2019 | This study |
| UMAF M1934 | MZ160949 | Mango | December 2019 | This study |
| UMAF M1935 | MZ160950 | Mango | December 2019 | This study |
| UMAF M1936 * | MZ160951 | Mango | December 2019 | This study |
| UMAF M1939 * | MZ160954 | Mango | December 2019 | This study |
| UMAF M1940 | MZ160955 | Mango | December 2019 | This study |
| UMAF M1941 | MZ160956 | Mango | January 2020 | This study |
| UMAF M1942 * | MZ160957 | Mango | January 2020 | This study |
| UMAF M1943 | MZ160958 | Mango | January 2020 | This study |
| UMAF M1944 | MZ160959 | Mango | January 2020 | This study |
| UMAF M1946 | MZ160961 | Avocado | February 2020 | This study |
| UMAF M1948 * | MZ160963 | Avocado | February 2020 | This study |
| UMAF M1950 | MZ160965 | Avocado | February 2020 | This study |
| UMAF M1951 | MZ160966 | Avocado | February 2020 | This study |
| UMAF M1953 * | MZ160968 | Avocado | February 2020 | This study |
| UMAF M1957 * | MZ160972 | Avocado | February 2020 | This study |
| UMAF M1962 | MZ160977 | Avocado | February 2020 | This study |
| UMAF M1965 | MZ160980 | Avocado | May 2020 | This study |
| UMAF M1969 | MZ160984 | Avocado | May 2020 | This study |
| UMAF M1973 | MZ160988 | Avocado | June 2020 | This study |
| UMAF M1975 * | MZ160990 | Avocado | June 2020 | This study |
| Genus *Neofusicoccum* | | | | |
| UMAF M1302 * | MZ160917 | Mango | February 2013 | This study |
| UMAF M1921 | MZ160936 | Mango | November 2019 | This study |
| UMAF M1927 | MZ160942 | Mango | November 2019 | This study |
| UMAF M1928 * | MZ160943 | Mango | December 2019 | This study |
| UMAF M1929 | MZ160944 | Mango | December 2019 | This study |
| UMAF M1937 * | MZ160952 | Mango | December 2019 | This study |
| UMAF M1938 * | MZ160953 | Mango | December 2019 | This study |
| UMAF M1945 * | MZ160960 | Avocado | February 2020 | This study |
| UMAF M1947 | MZ160962 | Avocado | February 2020 | This study |
| UMAF M1949 * | MZ160964 | Avocado | February 2020 | This study |
| UMAF M1960 | MZ160975 | Avocado | February 2020 | This study |
| UMAF M1961 * | MZ160976 | Avocado | February 2020 | This study |
| UMAF M1963 | MZ160978 | Avocado | May 2020 | This study |
| UMAF M1964 * | MZ160979 | Avocado | May 2020 | This study |
| UMAF M1966 | MZ160981 | Avocado | May 2020 | This study |

**Table 1.** *Cont.*

| Fungal Isolates Code and Accession Number | | Source Fruit | Date of Isolation | Reference |
|---|---|---|---|---|
| Genus *Stemphylium* | | | | |
| UMAF M1301 * | MZ160916 | Mango | February 2013 | This study |
| UMAF M1303 * | MZ160918 | Mango | February 2013 | This study |
| UMAF M1304 | MZ160919 | Mango | February 2013 | This study |
| UMAF M1305 | MZ160920 | Mango | February 2013 | This study |
| UMAF M1307 | MZ160922 | Mango | February 2013 | This study |
| UMAF M1308 | MZ160923 | Mango | February 2013 | This study |
| UMAF M1309 * | MZ160924 | Mango | February 2013 | This study |
| UMAF M1917 | MZ160932 | Mango | October 2019 | This study |
| UMAF M1919 * | MZ160934 | Mango | November 2019 | This study |
| UMAF M1930 | MZ160945 | Mango | December 2019 | This study |
| **Other species of fungus isolated from mango and avocado fruits** | | | | |
| Genus *Aureobasidium* | | | | |
| UMAF M1911 * | MZ160926 | Mango | October 2019 | This study |
| UMAF M1912 | MZ160927 | Mango | October 2019 | This study |
| UMAF M1914 | MZ160929 | Mango | October 2019 | This study |
| UMAF M1922 | MZ160937 | Mango | November 2019 | This study |
| UMAF M1923 * | MZ160938 | Mango | November 2019 | This study |
| Genus *Colletotrichum* | | | | |
| UMAF M1925 * | MZ160940 | Mango | November 2019 | This study |
| UMAF M1926 * | MZ160941 | Mango | November 2019 | This study |
| UMAF M1958 | MZ160973 | Avocado | February 2020 | This study |
| UMAF M1959 * | MZ160974 | Avocado | February 2020 | This study |
| Genus *Lasiodiplodia* | | | | |
| UMAF M1967 * | MZ160982 | Avocado | May 2020 | This study |
| UMAF M1972 * | MZ160987 | Avocado | June 2020 | This study |
| Genus *Nigrospora* | | | | |
| UMAF M1924 | MZ160939 | Mango | November 2019 | This study |
| UMAF M1952 * | MZ160967 | Avocado | February 2020 | This study |
| Genus *Trichoderma* | | | | |
| UMAF M1956 | MZ160971 | Avocado | February 2020 | This study |
| UMAF M1974 | MZ160989 | Avocado | June 2020 | This study |
| Genus *Fusarium* | | | | |
| UMAF M1968 * | MZ160983 | Avocado | May 2020 | This study |
| UMAF M1970 | MZ160985 | Avocado | May 2020 | This study |
| Other fungus isolated | | | | |
| *Botryosphaeria* sp. | | | | |
| UMAF M1920 * | MZ160935 | Mango | November 2019 | This study |
| *Pestalotiopsis* sp. | | | | |
| UMAF M1971 * | MZ160986 | Avocado | June 2020 | This study |
| *Rosellinia* sp. | | | | |
| UMAF M1954 | MZ160969 | Avocado | February 2020 | This study |
| *Xylaria* sp. | | | | |
| UMAF M1955 | MZ160970 | Avocado | February 2020 | This study |

(*) Fungal isolates from this study chosen for antagonism experiments.

Additionally, to check the presence of fungi associated with the avocado fruits, fungal presence on the avocado skin was tested. The avocado fruits for analysis were received from February to June 2020, and they did not show visible rot symptoms. For fungal isolation from the avocado fruit surface, a similar isolation procedure was performed as described above. Small portions from disinfected avocado skin were deposited on acidified PDA and incubated at 25 °C for one week, and the different fungi growing from the samples were isolated in pure culture for further identification.

From the fungal collection of 75 isolates, 34 of them were chosen as representations for further experiments. To do this, representatives of all genera were considered, choosing isolates from different batches of fruits and isolated at different times (Table 1, and Figure S2 Supplemental Material).

For experiments with biocontrol bacteria, fermented Bv was commercially produced and kindly provided by Koppert Biological Systems (Berkel en Rodenrijs, The Netherlands). The fermented bacterial-based suspension of Bv contained $10^9$ spores/mL and was maintained at 4 °C. For Pc, a 24 h culture in tryptone-peptone-glycerol (TPG) broth [23] at 25 °C with orbital shaking at 150 rpm was centrifuged at 10,000 rpm for 5 min at 4 °C, and the bacterial pellet was resuspended in sterile distilled water, rendering a bacterial suspension of approximately $10^8$ cfu/mL (0.8 of O.D. at 600 nm), This process was performed every time that a Pc suspension was needed. The concentration and viability of both products were monitored before each experimental treatment.

## 2.2. Molecular Identification of Fungal Isolates

To assign an initial identification of the fungal collection members, the sequence of the internal transcribed spacer (ITS) rRNA was obtained [24] and compared with those available at the database of the National Center for Biotechnology Information webpage (NCBI). The partial ITS sequences were already present in the database of the main fungal groups isolated (*Neofusicoccum* spp., *Alternaria* spp. and *Stemphylium* spp.). To identify the species of *Neofusicoccum* isolates used in further experiments, an additional sequence of β-tubulin was studied [25].

## 2.3. Reproduction of Rot Symptoms in Mango Fruit

To study the pathogenicity of the fungal isolates obtained, experimental infections were performed, and the reproduction of rot symptoms on mango fruit was reported. For that, six asymptomatic mango fruits (vars. Osteen) completely developed and ready for harvest but still not ripened for consumption, were superficially disinfected and artificially inoculated with selected fungal isolates from the most frequently detected species. Specifically, seven isolates of *Alternaria* spp. (UMAF M1310, UMAF M1913, UMAF M1931, UMAF M1933, UMAF M1936, UMAF M1939, and UMAF M1942) and four of *Neofusicoccum* spp. (UMAF M1302, UMAF M1928, UMAF M1937, and UMAF M1938), were used.

For artificial inoculations, hyphal suspensions were used [26,27]. For this, 5 mL of sterilized distilled water was added to the fungal colony growing on a PDA plate at 25 °C for 5 days, and the mycelium was detached using a sterile scalpel. The obtained mycelial mass was then added to a new tube with 5 mL of sterilized distilled water and 0.25 g of sterile quartz sand, and vortexed for at least 1 min until the suspension appeared homogeneous. Then, the sand was removed, and the suspension was adjusted to 50 mL. These inocula were found to be within the range of $2 \times 10^3$ to $5 \times 10^3$ CFU mL$^{-1}$.

Ten microlitres of the hyphal suspension were used for inoculating each wound previously made in a disinfected mango fruit. The wounds were produced by pricking 0.5 cm deep using a sterile hypodermic needle BD Microlance$^{TM}$ (Madrid, Spain) 21G × 1½ "Nr. 2, (0.8 × 40 mm). The negative control was inoculated with 10 μL of sterilized distilled water. Every isolate tested was inoculated at six points (three points per mango fruit). The inoculated preclimateric fruits were then incubated at room temperature for 7 days. The presence of rotten or dark areas at the infection points was reported (Figure 1A–E). The experiment was repeated at least twice.

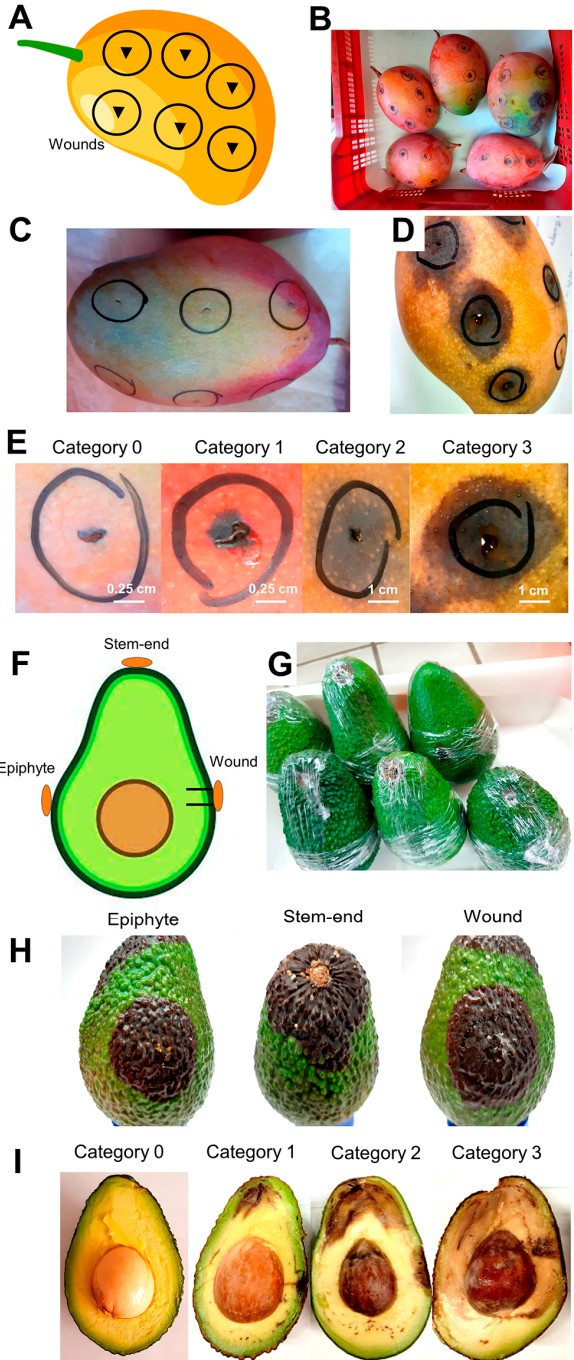

**Figure 1.** Inoculation procedure for rot symptom development in preliminary assays for symptom establishment. (**A**,**F**) Drawing that represents the strategies used in the inoculation of fungal pathogens in mango (**A**) and avocado (**F**). (**B**,**G**) Inoculated fruits ready for incubation at room temperature for seven days. (**C**) Mango fruit recently inoculated with fungal suspension. (**D**) Superficial rot symptoms were observed on mango skin after seven days of incubation at room temperature. (**E**) Categories of severity used in mango artificial infection assays. Category 0 means no symptoms, Category 1 includes a necrotic area less than 0.5 cm in diameter, Category 2 includes a necrotic area from 0.5 to 2 cm in diameter, and Category 3 includes a necrotic area more than 2 cm in diameter. (**H**) Superficial rot symptoms observed on avocado skin after eight days of incubation at room temperature at the three different inoculation points. (**I**) Categories of severity used in avocado artificial inoculated assays. Category 0 means no symptoms, Category 1 includes less than 25% of the necrotic area, Category 2 includes 25% of the necrotic area, and Category 3 includes more than 25% of the necrotic area.

### 2.4. Reproduction of Rot Symptoms in Avocado Fruits

To determine whether the fungi isolated from symptomatic mango fruits could also produce rot symptoms in avocado fruits, different wild-type fungal representatives were artificially inoculated on asymptomatic avocado fruits. Six fruits of avocados were used per isolate tested (var. Fuerte n = 3, var. Hass n = 3). The fruits were completely developed and ready for harvest but still not ripened for consumption. Five isolates of *Neofusicoccum* spp. (UMAF M1302, UMAF M1937, UMAF M1938, UMAF M1960, and UMAF M1964), and four isolates of *Alternaria* spp. (UMAF M1931, UMAF M1948, UMAF M1942, and UMAF M1969) were inoculated. Three different inoculation procedures were tested: (i) by infecting the fruit stem end, (ii) by inoculation on wounds provoked through the skin, and (iii) by direct contact as epiphyte with the unaltered avocado skin (Figure 1F–I). To ensure infection of fungal isolates, a PDA disc from the edge of 5-day-old mycelial growth of the test fungus was deposited onto the superficially disinfected fruit surface and fixed by Parafilm [28,29]. The inoculated avocados were then incubated at room temperature for 8 days, and the appearance of rot symptoms outside and inside the fruit was recorded. The experiment was repeated at least twice.

### 2.5. Antifungal Features of the Biocontrol Bacteria Used in Dual Cultures In Vitro

The antagonism of the biocontrol strains Bv UMAF6639 and Pc PCL1606 against selected representatives of the fungal collection (Table 1) was determined in dual cultures in vitro [30], and the inhibition produced by the bacteria was quantified as the percentage of fungal growth in bacteria presence and it was calculated as follows:

$$\text{Growth percentage} = 100 - [(d/D) \times 100]$$

D = the distance in mm between the seeding point of the fungus and the bacteria (20 mm), and d = the average inhibition distance in mm between the edge of fungal growth and the edge of the bacterial colony after incubation (Figure S3, Supplemental Materials).

For that, bacteria and fungi were plated at the same time on a PDA medium. The assay was performed three times with three internal replicas each. Additionally, nonantagonist bacteria and plates without antagonist were used as controls to test the effect of TPG medium on fungal growth.

Additionally, the effect of volatile organic compounds (VOCs), produced by antagonist bacteria on the growth of fungal pathogens, was also analysed to determine their potential inhibitory ability, by following the Arrebola et al. [31] procedure (Figure S3, Supplemental Materials). Before each assay, both antagonistic bacteria were grown in fresh LB plates for 24 h at 25 °C. For the assay, the bacteria were spread onto the surface of the plate using a sterile swab, Bv was cultured in LB plates, and Pc in TPG plates. For testing bacterial antagonist and pathogenic fungus combinations, a Petri dish sandwich was made, and the top Petri dish was composed of a Petri dish bacterial culture (lower plate). Thereafter, a small piece of PDA plate with mycelial growth was placed in the centre of the bottom PDA plate (upper plate). The upper and lower plates were sealed together with Parafilm and incubated at 25 °C for 8 days. The area of fungal growth was measured using the Quantity 1-D analysis software version 4.6.6 PC (Bio-Rad Laboratories, Inc., Madrid, Spain). Each fungal isolate test was repeated five times with two internal replicates each.

### 2.6. Evaluation of Experimental Biocontrol Treatments on Mango Fruits

The effectiveness of experimental treatments with biocontrol bacteria on artificially in-oculated mangos fruits was analysed. Three varieties of mango were analysed, var. Osteen, Kent and Keitt. For each variety, seventy-four fruits were used, that were distributed in the different treatment combinations (preventive/curative, Bacillus/Pseudomonas, three fungal isolates inoculated and controls). Every combination was analysed by six independents fruits with six inoculation points each [21].

The preclimaterically harvested fruits were washed and superficially disinfected with 70% ethanol, the fungal isolates were inoculated as previously described, and the bacterial treatment was applied 24 h after inoculation for the curative test, or 24 h before inoculation for the preventive test (as explained below). The assays were carried out from September to November according to mango variety season.

To test a preventive approach, wounds were made on the superficially disinfected mango fruit as previously described. Then, the bacterial application was applied onto the mango fruits, prior to application of the fungal hyphal suspension. For Bv treatment, 250 µL of the fermented bacterial suspension was added to 250 mL of sterile water to a final concentration of $10^6$ CFU mL$^{-1}$ of bacteria. For Pc, an overnight culture in TPG broth at 25 °C was adjusted to 0.8 (O.D. at 600 nm, corresponding to $10^8$ CFU mL$^{-1}$), and 2.5 mL from this culture was used to prepare a final 250 mL of Pc treatment ($10^6$ CFU mL$^{-1}$). The bacterial suspension was amended with Nu-Film® (1 cm$^3$ L$^{-1}$) as an adherent adjuvant used for applications in ecological agriculture. The bacterial suspension was applied by spraying onto the disinfected fruits until they were completely wet and saturated with antagonist.

After an incubation period of 24 h at room temperature, the mango fruits were then inoculated with the fungal pathogens at the wounds previously made as described previously. Three different pathogenic fungal isolates were used: two isolates of *Neofusiccocum parvum* (UMAF M1302 and UMAF M1937) and one of *Neofusicoccum mediterraneum* (UMAF M1938).

For testing a curative treatment, the experimental design was very similar to that of the preventive treatment, but fungal infection was performed after the fruit disinfection, and followed, 24 h later, by the bacterial application.

Three different control treatments were performed to properly evaluate symptom development. The infection controls were mango fruits inoculated with fungi but treated with sterile distillate water (with adjuvant) instead of the bacterial treatment. A second control was fruit treated only with the bacterial application and sterile distillate water with the adjuvant instead of the fungal inoculation for monitoring the bacterial population without fungal presence. Finally, a negative control was performed using wounded mango water inoculated and then treated only with sterile distillate water with adjuvant to monitor the fruit ripening under assay conditions.

The inoculated fruits were maintained at room temperature for 9 days with natural HR (ranging from 55 to 75%), and a light–dark cycle (12–12 h). Then, symptom incidence was monitored as the presence/absence of rot symptoms at the inoculation points. To report the severity levels at the inoculation point, a symptom scale was used: category 0 (no visible symptoms), category 1 (from 0 to 0.5 cm of necrotic area diameter), category 2 (from 0.5 to 2 cm of necrotic area diameter), and category 3 (more than 2 cm of necrotic area diameter) (Figure 1E).

To fulfil Koch's postulates, reisolation and characterization of the rot symptoms were performed following the procedures previously described to confirm the presence of inoculated fungi associated with the corresponding rot symptoms.

Simultaneously, bacterial counts were monitored during the experiments, sampling 1 cm$^2$ on the treated inoculation point, and homogenized in 1 mL of sterile water, 10-fold diluted, and 100 µL of samples was cultured on (i) Pseudomonas-selective medium (Pseudomonas Isolation Agar—Sigma Aldrich, Burlington, MA, USA) plates for Pc counts, and (ii) LB for Bv counts. All media were supplemented with cycloheximide (100 µg mL$^{-1}$) to avoid fungal growth. The plates were incubated at 25 °C for 24 h for Pc and 37 °C 24 h for Bv and colonies were counted for calculation. To obtain the Bv spore counts, the dilution samples were heated at 80 °C for 10 min prior to being plated.

*2.7. Fungal Pathogenicity and Experimental Biocontrol on Avocado Fruits*

Nine fruits of Hass avocados, previously washed and superficially disinfected with 70% ethanol, were used to test the pathogenicity of every combination of selected fungal isolates, bacterial treatment, and preventive/curative application. One hundred forty-four av-

ocado fruits were used in total for each independent experiment, which was performed from March to April, and was repeated twice.

The inoculum of the pathogenic fungal isolates *Neofusicoccum* spp. (UMAF M1937, UMAF M1928, and UMAF M1945) were prepared as previously described; however, the total inoculum volume was reduced to 15 mL, and was applied in avocado by sterile brush at the stem end. The inoculated avocado fruits were maintained at room temperature for 13 days with natural HR and a light–dark light cycle as previously mentioned. The bacterial treatments were applied as previously described for mango trials, and the symptom incidence and severity, as well as bacterial counts during experiments were monitored. For symptom evaluation, it was necessary to open the fruits longitudinally to see the internal rots. In this case, three fruits per independent fungal pathogen were processed at each sampling point (Figure 1I). Severity levels were ranked using the following symptom scale: Category 0 (no symptoms), Category 1 (less than 25% of necrotic area), Category 2 (25% of necrotic area), and Category 3 (more than 25% of necrotic area) (Figure 1I).

### 2.8. Bacterial Survival on Commercial Fruit under Open Field Conditions

To check biocontrol bacteria (Pc and Bv) survival on the mango and avocado fruit surface under commercial conditions, open field trials were carried out following a preventive approach. Three individual trees with preclimateric fruits (ready for harvest) were treated with fermented Bv ($10^9$ CFU mL$^{-1}$) at 10 mL L$^{-1}$ and Nu-Film (1 mL L$^{-1}$), three other trees were treated with Pc cells ($10^9$ CFU mL$^{-1}$) at 10 mL L$^{-1}$ and Nu-Film (1 mL L$^{-1}$), and three additional trees were used as controls, which were treated only with water + Nu-Film (1 mL L$^{-1}$). Five to six litres of treatment were applied per tree to ensure coverage of the canopy with the foliar treatment. Bacterial counts were obtained from two randomly picked fruits from each of the three treated trees under the same treatment. Two sampling times were performed for mangoes, on Day 1 and Day 15 postinoculation, and four sampling times were performed for avocado, 1, 8, 15, and 22 days postinoculation. The mango tree trials were performed in November 2019 and the avocado trials were performed in February 2020. The average temperatures were a maximum temperature of 24 °C and a minimum temperature of 15 °C in November 2019 and a maximum of 17 °C and a minimum of 8 °C in February 2020.

For bacterial counts from mango and avocado fruits, an area of approximately 60 cm$^2$ of the fruit surface was rubbed with a sterile swab. Then, the cotton pit was cut and immersed in 2 mL of saline solution (0.85% of NaCl) and vortexed for 1 min, and then 10-fold serial dilutions on saline solution were performed. The samples of Bv were cultured in LB supplemented with cycloheximide (100 µg mL$^{-1}$) and incubated at 37 °C for 24 h. For the *Bacillus* spore count, the 10-fold dilutions were heated at 80 °C for 10 min, cultured in LB supplemented with cycloheximide, and incubated at 37 °C for 24 h. For the Pc counts, the samples were plated on Pseudomonas Isolation Agar (Sigma-Aldrich, Burlington, MA, USA) and incubated at 25 °C for 48 h. The Pc and Bv colonies were confirmed by detection using a specific PCR reaction [32,33].

### 2.9. Statistical Analysis

For the antagonism assays, the data distributions were tested using one-way analysis of variance (ANOVA) followed by Fisher's least significant difference test with Bonferroni's correction ($p = 0.05$). For the incidence and severity assays, significant differences between different bacterial treatments were analysed using an unpaired Student's *t*-test ($p = 0.05$). All data analyses were performed using IBM SPSS statistics 25 software (SPSS, Inc., Chicago, IL, USA). The average of the measures has been considered as representative results in the figures corresponding to experiments with repetitions.

## 3. Results

### 3.1. Collection of Fungal Isolates Obtained from Mango and Avocado Fruits

Commercial mango fruits with evident symptoms of rottenness were received from October 2019 to January 2020. These symptoms consisted of extensive dark spots on the fruit surface and at the stem end and the softening of the pulp (Figure S1). A total of forty-four fungal isolates were obtained from these rot symptoms on mango, including a collection of 10 other fungi isolates previously obtained from similar rot symptoms in mango in 2013. On the other hand, a collection of 31 fungal isolates was obtained from the skin of asymptomatic avocado fruits collected from the same geographical area where mango grows. Thus, a fungal collection of 75 isolates was obtained from mango and avocado fruits and analysed in this study (Table 1). Their identification was proposed after ITS sequence analysis and comparison in the NCBI database (Table S1, Supplemental Materials). Seventy-two percent of the analysed isolates belonged to three fungal genera, *Alternaria*, *Neofusicoccum*, and *Stemphylium*; however, most of the *Stemphylium* sp. isolates belonged to those strains previously obtained in 2013. Therefore, *Alternaria* spp. and *Neofusicoccum* spp. were the main fungi isolated in the current study, accounting 38.7% and 20%, respectively. Other species found in mango and avocado fruit skin were *Aureobasidium* sp. and *Colletotrichum* sp. at 6.7% and 5.3%, respectively. The rest of the isolates obtained in this study belonged to common species that can be easily found in agricultural fields (Table 1).

### 3.2. Identification of the Causal Agent of Postharvest Fruit Rot

With the objective of identifying the etiological agent of mango fruit rot from the fungi previously isolated from mango and avocado fruits, experimental infections with selected isolates of the main fungal species were performed. Isolates of *Alternaria* spp. and *Neofusicoccum* spp. were used to reproduce mango fruit rot symptoms. The isolates assigned to the *Alternaria* genus assayed in this work were not able to cause rot symptoms on mango. However, the selected isolates of *Neofusicoccum* spp. tested developed rot symptoms similar those observed on commercial mangoes with rot diseases (Figure 1C–E; Table 2); characterized by extensive dark spots with pulp softening and no sunken skin, similar to the mango samples received from storage.

The study of potential cross-infections in avocado among the fungal isolates obtained from mango revealed that representatives of the two main fungal genera (*Alternaria* and *Neofusicoccum*) were able to produce rot symptoms, but the isolates belonging to *Neofusic-occum* spp. showed higher incidence on avocado fruits, with necrotic rots at the different inoculation points that could easily be seen (Figure 1). When the infected avocados were cut and opened, several rot symptoms were observed inside, with the prevailing rot coming from the stem end as the point of inoculation. On the other hand, isolates belonging to the genus *Alternaria* also showed rot symptoms in avocado fruits but were smaller in size than those produced by *Neofusicoccum* sp. and were usually not visible on the outside of the avocado skin (Figure 1). At this point in the study, several isolates of *Neofusicoccum* spp. were confirmed by sequencing the partial CDS of β-tubulin (Table 3) for further experiments. According to ITS and β-tubulin identification, isolates UMAF M1302, UMAF M1928, and UMAF M1937 were identified as *Neofusiccocum parvum*, and UMAF M1938 and UMAF M1945 were identified as *N. mediterraneum*. The species of isolate UMAF M1961 was not confirmed due to a discrepancy between ITS and β-tubulin sequence analyses (Tables 3 and S1).

**Table 2.** Rot production assay in mango and avocado fruits with fungal species isolated from fruit skin. One variety of mango (Osteen) and two cultivars of avocados (Fuerte and Hass) were used. Three inoculation points, stem end, artificial wound, and direct contact with the fruit skin, which was considered epiphyte, were tested in avocado. The results were obtained from two independent experiments with three replicas each (six inoculated points). The results show the number of inoculated points with rot symptoms relating to the total number of inoculated points.

| Genus | Isolate | Sc | Mango | Avocado | | | | | |
|---|---|---|---|---|---|---|---|---|---|
| | | | Osteen | Fuerte | | | Hass | | |
| | | | | P | W | E | P | W | E |
| *Neofusic.* | (Np) UMAF M1302 | M | 3/6 | 4/6 | 5/6 | 3/6 | 4/6 | 3/6 | 3/6 |
| | (Np) UMAF M1928 | M | 4/6 | nd | nd | nd | nd | nd | nd |
| | (Np) UMAF M1937 | M | 6/6 | 5/6 | 4/6 | 3/6 | 4/6 | 3/6 | 3/6 |
| | (Nm) UMAF M1938 | M | 4/6 | 4/6 | 4/6 | 3/6 | 3/6 | 3/6 | 3/6 |
| | UMAF M1960 | A | nd | 5/6 | 5/6 | 3/6 | 5/6 | 3/6 | 3/6 |
| | UMAF M1964 | A | nd | 5/6 | 3/6 | 3/6 | 6/6 | 3/6 | 3/6 |
| *Alternaria* | UMAF M1310 | M | 0/6 | nd | nd | nd | nd | nd | nd |
| | UMAF M1913 | M | 0/6 | nd | nd | nd | nd | nd | nd |
| | UMAF M1931 | M | 0/6 | 1/6 | 1/6 | 0/6 | 2/6 | 1/6 | 0/6 |
| | UMAF M1939 | M | 0/6 | nd | nd | nd | nd | nd | nd |
| | UMAF M1933 | M | 0/6 | nd | nd | nd | nd | nd | nd |
| | UMAF M1936 | M | 0/6 | nd | nd | nd | nd | nd | nd |
| | UMAF M1942 | M | 0/6 | 4/6 | 4/6 | 3/6 | 1/6 | 0/6 | 0/6 |
| | UMAF M1948 | A | nd | 2/6 | 0/6 | 0/6 | 2/6 | 0/6 | 0/6 |
| | UMAF M1969 | A | nd | 3/6 | 3/6 | 3/6 | 0/6 | 0/6 | 0/6 |
| Control | Sterile water | | 0/6 | 0/6 | 0/6 | 0/6 | 0/6 | 0/6 | 0/6 |

| | |
|---|---|
| 0/6 | 0 infected point from 6 inoculated points |
| 1/6 | 1 infected point from 6 inoculated points |
| 2/6 | 2 infected point from 6 inoculated points |
| 3/6 | 3 infected point from 6 inoculated points |
| 4/6 | 4 infected point from 6 inoculated points |
| 5/6 | 5 infected point from 6 inoculated points |
| 6/6 | 6 infected point from 6 inoculated points |

Sc: source, M: mango skin fruit, A: avocado skin fruit. P: peduncle, W: wound, E: epiphyte, nd: no data (not assayed). *Neofusic.*: genus *Neofusicoccum*; Np: *Neofusicoccum parvum*.

**Table 3.** Specie identification of the main isolates used in the study.

| Fungal Strain | Accession Numbers | Product | Subject Organism |
|---|---|---|---|
| UMAF M1302 | MZ160917 | ITS | *Neofusicoccum parvum* |
| | OR463039 | β-Tubulin | *Neofusicoccum parvum* |
| UMAF M1928 | MN160943 | ITS | *Neofusicoccum parvum* |
| | OR463040 | β-Tubulin | *Neofusicoccum parvum* |
| UMAF M1937 | MN160952 | ITS | *Neofusicoccum parvum* |
| | OR463041 | β-Tubulin | *Neofusicoccum parvum* |
| UMAF M1938 | MN160953 | ITS | *Neofusicoccum mediterraneum* |
| | OR463042 | β-Tubulin | *Neofusicoccum mediterraneum* |
| UMAF M1945 | MN160960 | ITS | *Neofusicoccum mediterraneum* |
| | OR463043 | β-Tubulin | *Neofusicoccum mediterraneum* |
| UMAF M1961 | MN160976 | ITS | *Neofusicoccum australe* |
| | OR463044 | β-Tubulin | *Neofusicoccum cryptoaustrale* |

### 3.3. Antagonism Assays of Biocontrol Agents Bacillus velezensis and Pseudomonas chlororaphis against Fungal Isolates

A selection of 34 representative fungal isolates from the obtained fungal collection (Table 1) was tested against the two antagonistic bacteria, *Bacillus velezensis* (Bv) strain UMAF6639 and *Pseudomonas chlororaphis* (Pc) strain PCL1606. The dual cultures in vitro (Figures S3 and S4) showed a general reduction in fungal growth in the presence of Bv or Pc, reporting significant differences in the inhibition values between Bv and Pc in most cases. Among the fungi analysed, those belonging to the *Neofusicoccum* genus, showed more tolerance to Pc presence (from 0 to 20% of inhibition); however, the fungal growth was inhibited by approximately 60% in the presence of Bv (Figure 2A). The inhibition of *Alternaria* spp. by Bv and Pc was similar except for some individual isolates. For the rest of the fungal isolates, the antagonism of Pc and Bv was dependent on the species but, in general, Bv displayed higher fungal inhibition than Pc, in many cases with significant differences (Figure S5A–C, Supplemental Materials).

Moreover, fungal inhibition by the production of volatile organic compounds (VOCs), also showed that both Bv and Pc inhibited fungal growth (Figures S3 and S4). In general, the VOCs of Pc inhibited the fungal growth more strongly than those of Bv, in many cases with significant differences. The genera clearly affected by bacterial VOCs were *Alternaria*, *Stemphylium*, and *Neofusicoccum*, with *Alternaria* and *Stemphylium* as the most-affected genera, with equal to or less than 25% of growth (Figure S5D–F, Supplemental Materials). In this sense, the growth of some *Neofusicoccum* spp. isolates reached 90% (isolates UMAF M1949 or UMAF M1964), thus reporting *Neofusicoccum* as the genus more tolerant to bacterial VOCs (Figure 2B).

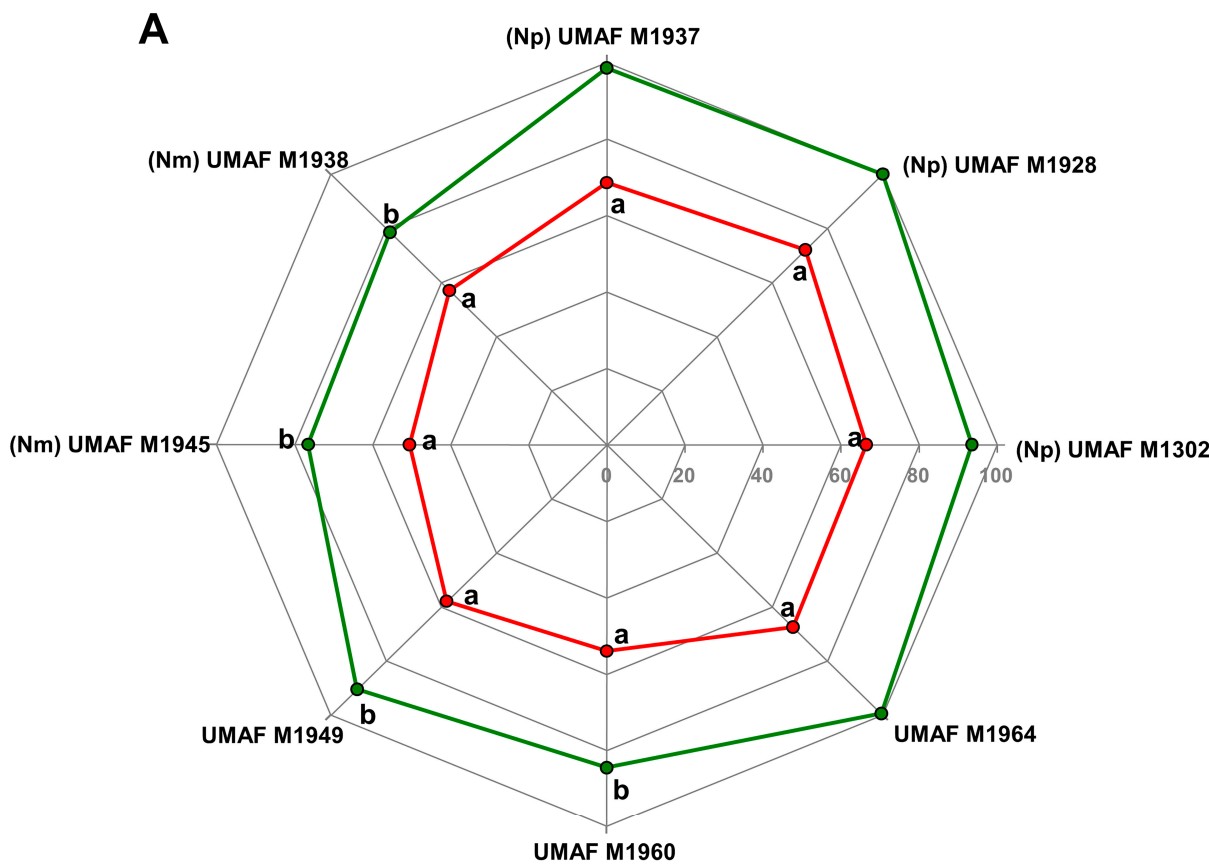

**Figure 2.** *Cont.*

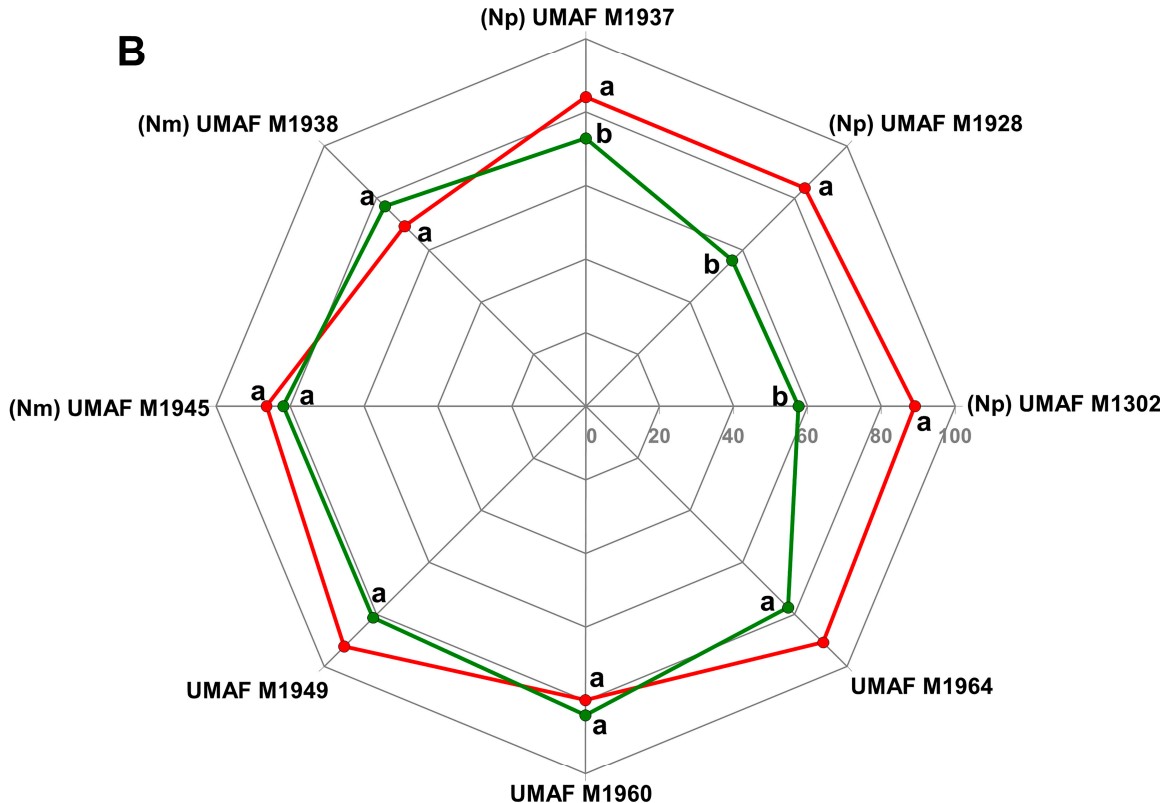

**Figure 2.** Radar representation of the percentage of growth (0–100%) of undetermined *Neofusicoccum* spp., *Neofusicoccum parvum* (Np), and *Neofusicoccum mediterraneum* (Nm), isolated in the current study against antagonist bacteria *Bacillus velezensis* UMAF6639 fermented in red and *Pseudomonas chlororaphis* PCL1606 in green. Values with letters show statistical significance in comparison with the control without bacteria. Values with different letters show significant differences between *Bacillus* and *Pseudomonas* levels of action relative to the control. (**A**) Percentage of growth of isolates by the dual technique for the antagonism test. (**B**) Percentage of growth of isolates incubated in front of antagonist bacteria for volatile organic compound analysis. The statistical analysis was conducted by one-way ANOVA using SPSS software 11.0.

### 3.4. Experimental Biocontrol Approaches with Bacterial Treatments

Biocontrol experiments on artificially infected fruits using antagonistic biocontrol bacteria were performed. The pathogenic fungal isolates used in these mango trials were the mango isolates of *Neofusicoccum parvum* UMAF M1302 (isolated in 2013) and UMAF M1937 and *N. mediterraneum* UMAF M1938, both isolated in the current study. For avocado trials *Neofusicoccum* sp. UMAF M1964 and UMAF M1960, isolated from avocado in the current study, and the isolate of *N. parvum* UMAF M1937 were used to check for cross-infections.

In general, the disease incidence was lower after preventive treatments than under curative treatments. Treatments with both fermented Bv UMAF6639 and Pc PCL1606 showed a decreased incidence of fruit rots in comparison with the control treatment, and this was more evident when using mango fruits than when using avocado fruits (Figure 3). Specifically, the preventive treatment with either of both biocontrol bacteria showed a significant difference from the control treatment over time in the mango assay. In the avocado assay, only Bv significantly decreased the incidence, but this could only be appreciated at the fifth day postinoculation. Otherwise, the curative treatments with bacteria did not show any differences from the control in either mango or avocado experiments (Figure 3).

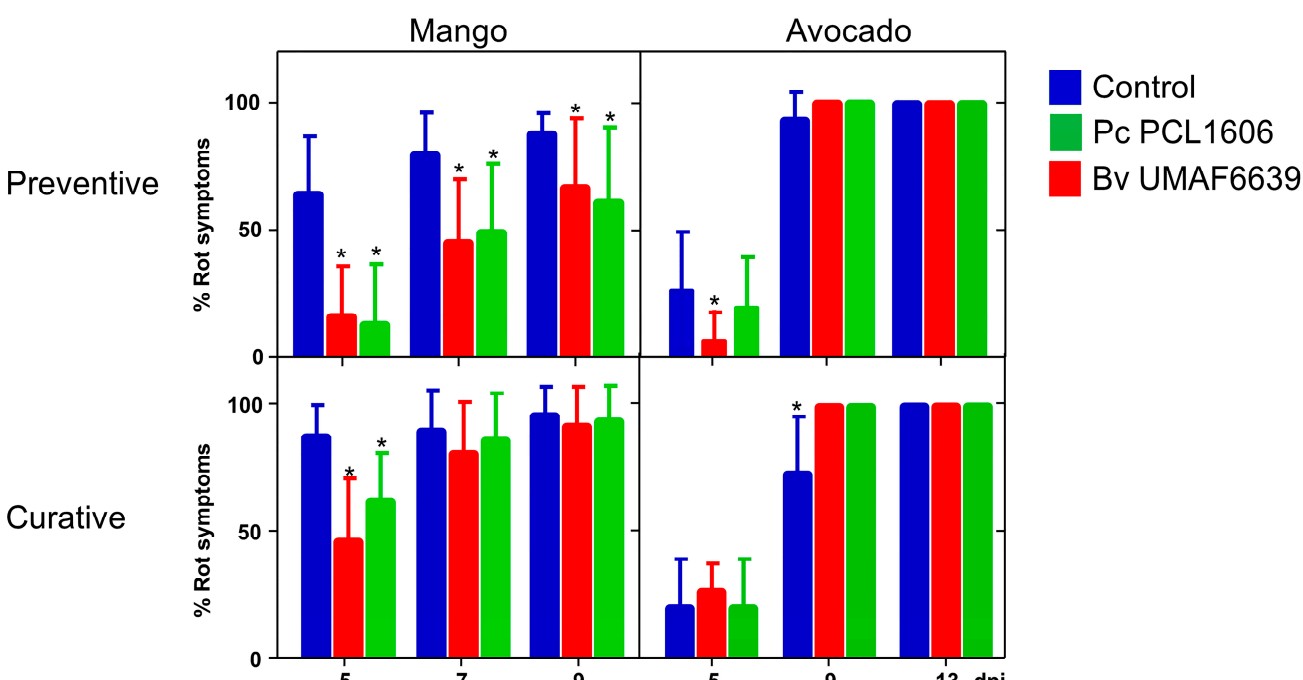

**Figure 3.** Percentage of rot incidence in the artificial infection experiment in mango and avocado fruits treated with antagonist bacteria *Pseudomonas chlororaphis* PCL1606 (green) and fermented *Bacillus velezensis* UMAF6639 (red). Sterile water with 1 mL L$^{-1}$ of adjuvant was used as a treatment control (blue). The cumulative incidence is analysed in preventive and curative treatment applications. Significant differences in the treatment compared to the control are marked with asterisks. The statistical analysis was conducted by Student's *t*-test using SPSS software 11.0.

Regarding the severity index displayed during both mango and avocado fruit trials, the results obtained depended on the fungal isolates used for artificial infections. Additionally, three varieties of mango fruits (Osteen, Kent and Keitt) were assayed to study the possible influence of mango variety on the severity of rot symptoms. In general, a decrease in severity in those fruits treated with Bv and Pc under preventive treatments was observed when compared to curative and control treatments. The severity of symptoms observed for the different fungal isolates tested in mango trials did not show significant differences in the variety Osteen in either preventive or curative treatment or with the two different bacteria applied (Figure 4). However, the severity displayed by the isolate *N. parvum* UMAF M1302 seems to be lower than that of the other two inoculated fungal isolates. This was more evident in the curative treatment, but without significant differences (Figure 4).

A comparison among the three mango fruit varieties tested at the fifth day postinocula­tion confirmed a statistically increased susceptibility to fungal rot development in the Keitt mango fruits during preventive treatments with biocontrol bacteria (Figure S6, Supplemen­tal Materials). Regarding the curative treatments, significant differences were shown by variety Osteen in Pc and Bv application, but this variety was generally more tolerant to fungal rot in comparison with Kent and Keitt (Figure S6).

The severity data from the avocado trial did not show differences among fungal isolates, displaying a very similar symptom evolution throughout the trials. Furthermore, the symptom severity shown by inoculated avocados of the Hass variety at the fifth day postinoculation only showed significant differences when Bv was used in the preventive application. The rest of the treatments did not show differences from the control.

# Mango (Osteen)

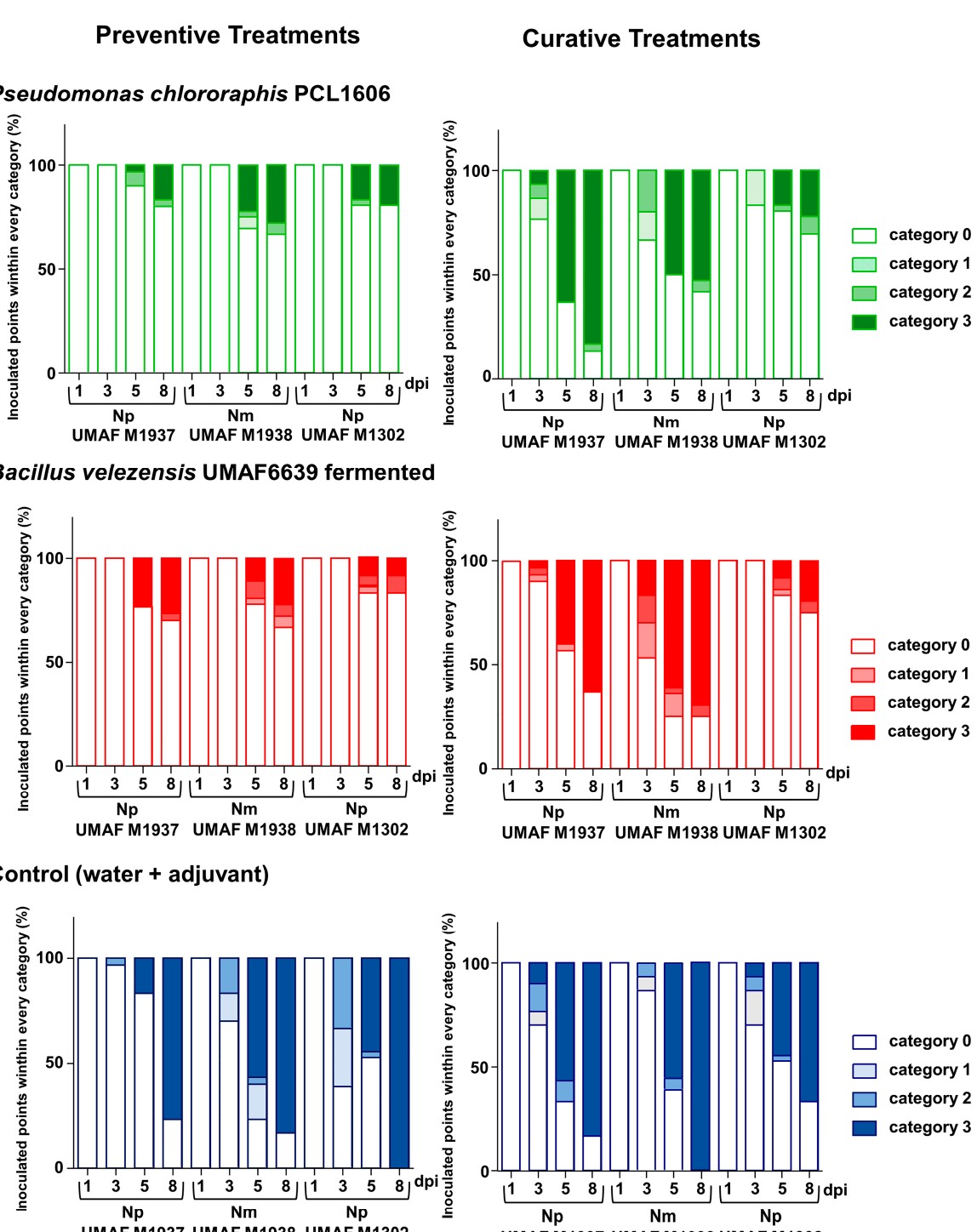

**Figure 4.** Severity symptom evolution on mango variety Osteen for eight days caused by three different fungi, *Neofusicoccum parvum* (UMAF M1937 and UMAF M1302) and *N. mediterraneum* (UMAF M1938). The symptom severity was analysed in fruits treated with *Pseudomonas chlororaphis* PCL1606 at a dose $10^6$ CFU mL$^{-1}$ with an adjuvant of 1 mL L$^{-1}$ (in green) and fermented *Bacillus velezensis* UMAF6639 at a dose $10^6$ CFU mL$^{-1}$ with an adjuvant of 1 mL L$^{-1}$ (in red). Sterile water with an adjuvant of 1 mL L$^{-1}$ was used as control (in blue). The treatments were applied in curative and preventive modes for analysis. The percentage of inoculated points belonging to each category of severity in preventive and curative treatment are represented by colour intensity.

To test the role of the bacterial persistence on fruits at different protection levels, bacterial counts were monitored in both mango and avocado fruits under preventive and curative treatments. The infected and control fruits without fungal infection were used to check the influence of the fungus on bacterial population counts (Figure 5). In general, the Pc treatment showed fluctuations in the population persistence during the assay, which were more pronounced in the curative than in the preventive treatments, but no significant reductions in the population were seen during the experiment. In fact, while the trend of the Pc counts in mango seemed to be able to maintain the initial levels, in avocado the population seems to slightly increase from the fifth day. Likewise, the Pc counts showed no significant differences between infected and noninfected mango fruits. In the case of Bv, the population showed stable levels of both spores and vegetative cells throughout the entire trial in both mango and avocado fruits and under both types of preventive and curative treatment applications (Figure 5).

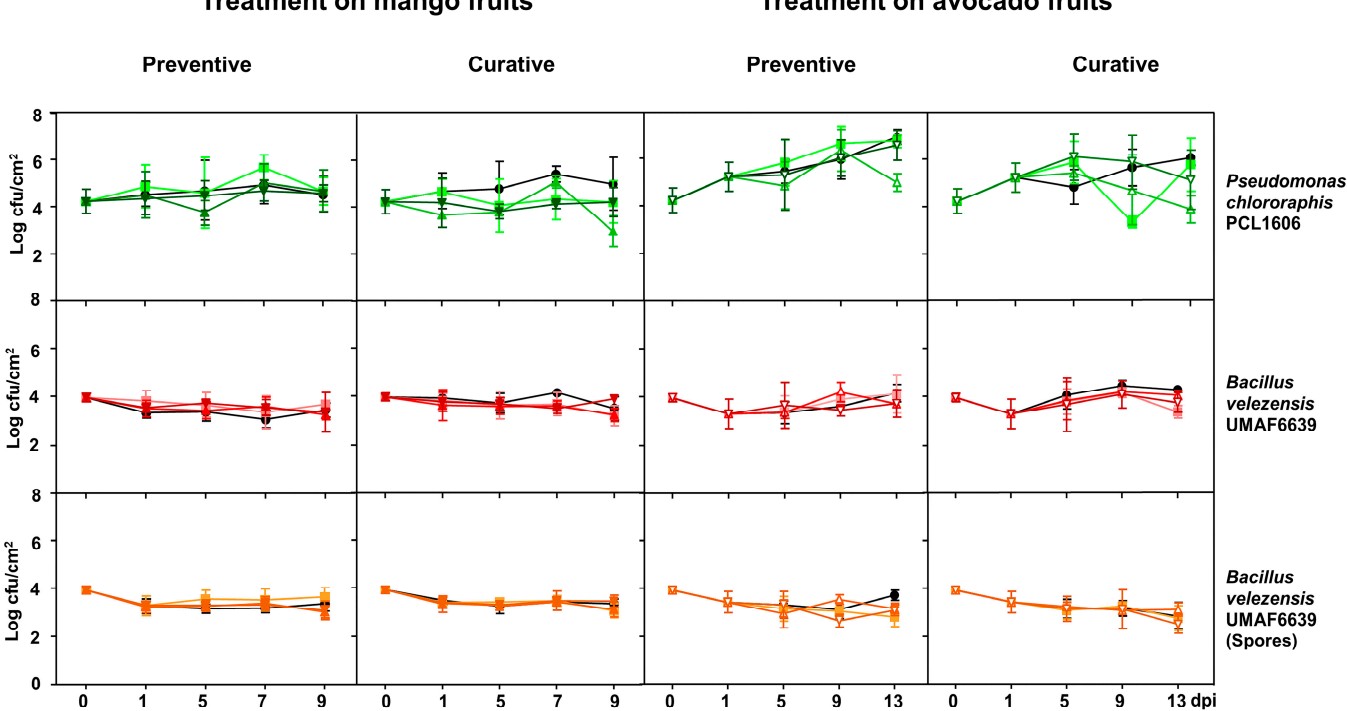

**Figure 5.** *Pseudomonas chlororaphis* PCL1606 (green scale) and *Bacillus velezensis* UMAF6639 fermented (vegetative cells in red, spores in orange) population counts on mango and avocado fruits inoculated with pathogen fungus and sterile water with adjuvant (1 mL L$^{-1}$) as a control. *Neofusicoccum parvum* (Np) UMAF M1937 (■), fungal inoculated in mango and avocado, Np UMAF M1302 (▼) only in mango and Np UMAF M1964 (▽) only in avocado. *N. mediterraneum* (Nm) UMAF M1938 (▲) only in mango and UMAF M1960 (△) only in avocado. The control is represented in black (●).

### 3.5. Viability of Bacterial Applications under Commercial Open Field Conditions

After confirmation of the potential interest of bacterial treatments to control fungal rot in detached mango and avocado fruits, a preventive application of bacterial biocontrol agents was selected for commercial trials under natural conditions. Bacterial survival after treatments with Bv and Pc suspensions was analysed, and the bacterial population was monitored (Figure 6). The results showed that after the preventive application of both biocontrol bacteria, a notable reduction in the amount of culturable bacteria on mango fruits was observed from the first day after application. Although the application of Pc was at an initial concentration of 10$^6$ CFU mL$^{-1}$, the detection levels during the whole assay were minimal. However, when Bv was applied, a decrease in the initial population was also observed, but its levels in both vegetative cells and spores were higher than those observed

for Pc. Remarkably, at 15 days post-treatment, the number of vegetative cells of Bv detected was more than double the number of spores (Figure 6). The preventive bacterial application on avocados showed an important decrease in the Pc population during the sampling time and, after 22 days, the Pc levels decreased four orders of magnitude. However, a minor reduction was observed in Bv levels, and after 22 days, the bacterial counts did not decrease below two orders of magnitude, maintaining the bacterial counts between $10^4$ and $10^5$ CFU per fruit, of vegetative cells and spores (Figure 6).

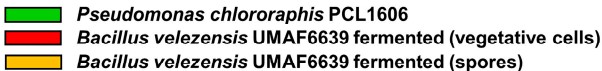

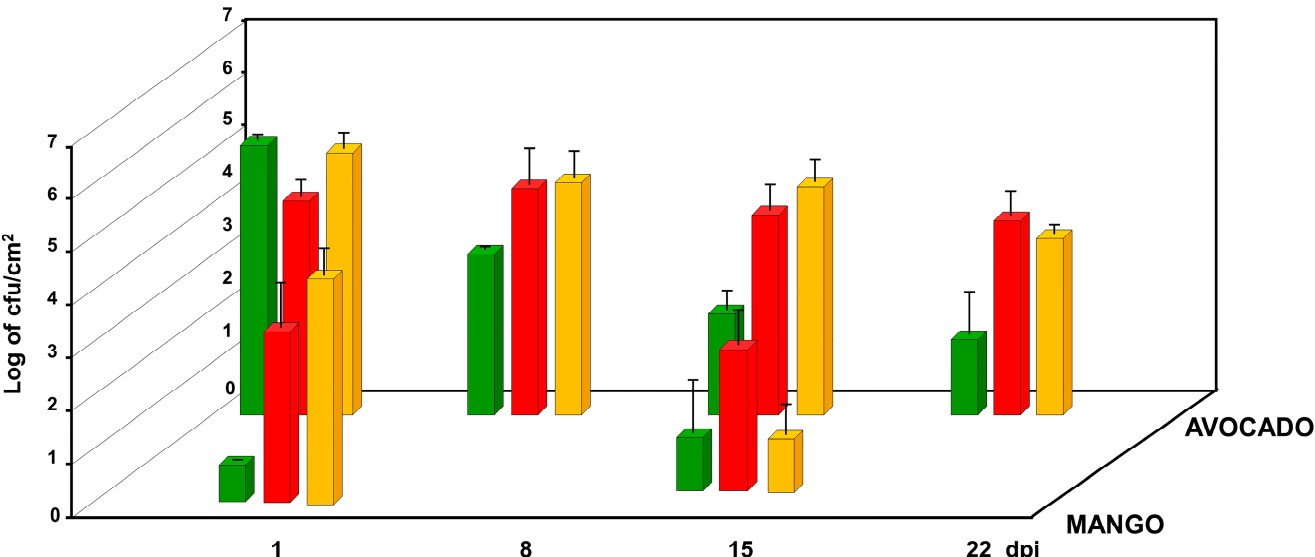

**Figure 6.** Bacterial counts obtained from fruits treated with preventive application in the field before harvest. Population of *Pseudomonas chlororaphis* PCL1606 (green) and *Bacillus velezensis* UMAF6639 (vegetative cells in red and spores in orange) after one day and fifteen days postinoculation (dpi) in mango fruits, and after one, eight, fifteen, and twenty-two days postinoculation (dpi) in avocado fruits.

## 4. Discussion

### 4.1. Identification of the Causal Agent of Mango Fruit Rot in Spain

In this study, symptomatic mango fruits that presented necrotic symptoms extending from the stem end of the fruit, spots on the fruit surface, and sometimes softening of the pulp (different from typical anthracnose) were mainly associated with the presence of the fungal genera *Alternaria* and *Neofusicoccum* (with two major species detected: *N. parvum* and *N. mediterraneum*). Both genera constitute part of the natural fungal community of agricultural crops, but their percentage could be increased on the fruit surface after a storage period [34]. Artificial infections revealed that the *Neofusicoccum parvum* and *N. mediterraneum* reproduced the observed natural mango rot symptoms, but rot symptoms were also detected in artificially inoculated avocado fruits, indicating the wide range of fruits in which these fungal species could cause potential rot diseases in stored fruits. *Neofusicoccum parvum* has been previously related to mango SER in Malaysia, affecting to Chok Anan, Waterlily, and Falan mango cultivars [35]. This current study confirms that additional commercial mango cultivars, Osteen, Kent, and Keitt can also be affected by *N. parvum*. On the other hand, the pathogenic species *N. mediterraneum* has previously been related to decay symptoms in *Vitis vinifera* [36] and is involved in branch dieback and shoot blight of several crops in Spain and California [37–39]. However, this fungus has been previously identified only as a causal agent of SER on Karuthakolumban mangoes in Sri Lanka [40]. This is the first report of *N. mediterraneum* as a causal agent of SER in other

different mango varieties and the first report of the presence of this potential pathogenic postharvest fungus in Spanish avocado fruits. However, *N. mediterraneum* has been detected as a causal agent of dieback in avocado trees by Arjona-Girona et al. [38] in Spanish crops.

The risk of potential cross infections of pathogenic fungi between mango and avocado fruits could be possible since the assayed isolates of *N. parvum* and *N. mediterraneum* from mango can reproduce rot symptoms in avocados after artificial infections. In fact, the isolation of very similar strains of *Neofusicoccum* sp. on asymptomatic avocado fruits can confirm the presence of these potential pathogens under field conditions simultaneously in both crops. This is a very well-known characteristic of these fungi, where the wide host range of *Neofusicoccum* species and the Botryosphaeriaceae family in general, combined with the ability to live on plants without visible symptoms for a long time, sometimes making them difficult to detect and control [9,34].

### 4.2. Experimental Biological Control of Stem-End Rot in Mango and Avocado Fruits

The biological control ability of antagonistic bacteria belonging to the *Bacillus* and *Pseudomonas* genera has been widely studied [41–45]. For this reason, the biocontrol strains *Bacillus velezensis* (Bv) UMAF6639 (formerly *B. amyloliquefaciens*) [20] and *Pseudomonas chlororaphis* (Pc) PCL1606 [18]. These agents are used for disease control on fruits and pose no health concerns for humans [46,47], and were assayed in the present study. The phyllospheric strain Bv UMAF6639 has been shown to produce powerful antifungal lipopeptides, such as iturin A, bacillomycin, and surfactin [20]. In the same way, the rhizospheric strain Pc PCL1606 produced a wide range of antifungal compounds, such as 2-hexyl, 5-propyl resorcinol, also known as HPR, pyrrolnitrin, and hydrogen cyanide, in addition to hydrolytic enzymes [19,48]. In general, both bacteria displayed antifungal characteristics against fungi after dual culture in vitro experiments. However, the genus *Neofusicoccum* seems to be tolerant of their inhibition.

On the other hand, these biocontrol strains have an additional mode of action such as competition for nutrients and space, actively colonizing the plant surface. Both *Bacillus* and *Pseudomonas* biocontrol strains have previously reported colonization and survival traits at the fruit surface [49–52]. Bv UMAF6639 populations, both vegetative and spores, remained stable during the fungal infection assay. This confirms the ability of the formulated Bv UMAF6639 spores to germinate on mango and avocado fruit surfaces, even though they are not a natural source of isolation, indicating that their efficacy could be extended beyond these phytopathogenic fungi. Moreover, the strain Bv UMAF6639 is already a good candidate in airborne biological control [20,53], similar to other commercial biological control treatments based on *Bacillus* spp. strains, e.g., Serenade® ASO (Bayer, Leverkusen, Germany). In contrast, Pc PCL1606 showed a fluctuant population on the fruit surface during this study. The *Pseudomonas* genus is unable to produce spores, which could make them more vulnerable to changes in humidity, temperature, and light. It is remarkable that Pc PCL1606 was initially isolated from avocado root [18], and it is naturally adapted to a dark, almost constant temperature, and more humid environment. Thus, when applied to fruits, incubated at room temperature, low light, and atmospheric levels of humidity, the challenging environment of the fruit surface could explain the recorded oscillations in the survival of the PCL1606 population.

Regarding the bacterial treatment application strategy (preventive or curative), a preventive application has been revealed to be more effective than a curative application, as previously described in other experimental models [54]. This could be due to a better colonization of infection points or wounds, thus preventing the entry of pathogenic fungi and adding this advantage to the antifungal activity exhibited [55].

### 4.3. Biocontrol Assays under Open Field Conditions

The fungal infection of postharvest rots usually originates in the crop field, and it is during ripening when the disease manifests, due to the complex hormonal changes [56]. For this reason, the preventive application of bacterial treatments on fully developed but not

yet harvested fruits were considered. The results showed that the survival of Pc PCL1606 on mango skin from commercial mango trees lasted not longer than one day after the bacterial application. The changes in temperature and humidity from day to night and solar radiation exposure could hinder the survival of these rhizobacteria. However, the phyllospheric strain Bv UMAF6639 was still detected after two weeks postapplication. Interestingly, when avocado trees were treated, Pc PCL1606 counts decreased but were still detected after 22 days postapplication, showing a better survival than that observed on the mango surface. In addition, the presence of Bv UMAF6639 was almost constant during the 22 days of the assay. Even when PcPCL1606 was isolated from avocado roots, this behaviour could be explained by the differential exposure of the mango and avocado fruits on the trees [57]. Mango fruits are exposed to the sun and the rest of the weather inclement; however, avocado fruits are usually protected under the tree canopy. Additionally, this result can also be influenced by the topography of the mango and avocado skin. The mango skin is completely smooth while the skin of the avocado presents roughness that helps to bind moisture and where the bacteria can take refuge [58].

## 5. Conclusions

According to our results, it is possible to establish that pathogenic *Neofusicoccum parvum* and *N. mediterraneum* are the main causes of mango postharvest rots in southern Spain. These fungi could also be potential pathogens to avocado fruits, suggesting a risk of cross-infections among crops. Biological strategies to control postharvest rots have shown better efficacy of preventive application strategies. Following the same strategy in open field experiments, Bv UMAF6639 was able to survive on the fruit surface longer than Pc PCL1606, suggesting that Bv UMAF6639 could constitute a possible treatment for mango and avocado crops.

**Supplementary Materials:** The following supporting information can be downloaded at: https://www.mdpi.com/article/10.3390/horticulturae10020166/s1, Figure S1: Mangos fruits with rot symptoms. The symptoms were developed during factory storage and analysed for causal agents in the current study; Figure S2: Graph of the numbers of isolated obtained from the rotten mangoes and their genera identification. Figure S3: Photograph illustrating the growth inhibition of fungal isolates caused by fermented *Bacillus velezensis* UMAF6639. Growth inhibition by antagonism in the dual test and by organic volatile compounds; Figure S4: Photograph illustrating the growth inhibition of fungal isolates caused by *Pseudomonas chlororaphis* PCL1606. Growth inhibition by antagonism in the dual test and by organic volatile compounds.; Figure S5: Radar representation of the percentage of growth of fungi isolated in the current study against antagonist bacteria, fermented *Bacillus velezensis* UMAF6639 (in red) and *Pseudomonas chlororaphis* PCL1606 (in green). Values with letters show statistical significance compared to the control without bacteria. Values with different letters show significant differences between bacterial levels of action. (A–C) Percentage of growth of isolates by dual technique for antagonism test: (A) isolates identified as *Alternaria* spp., (B) isolates identified as *Stemphylium* spp., (C) other fungal isolates obtained (Table 1). (D–F) Percentage of growth of isolates incubated in front of antagonist bacteria for volatile organic compound analysis: (D) isolates identified as *Alternaria* spp. axis scale up to 25%, (E) isolates identified as *Stemphylium* spp. axis scale up to 25%, (F) other fungal isolates obtained (Table 1). Statistical analysis was conducted by one-way ANOVA using SPSS software; Figure S6: Percentage of inoculated points with rot symptoms belonging to each category of severity obtained on the fifth day postinoculation of the three mango varieties: Osteen, Kent, and Keitt. The mangos fruits were inoculated with three fungal isolates, two *Neofusicoccum parvum* (UMAF M1937 and UMAF M1302) and one *N. mediterraneum* (UMAF M1938). The fruits were subjected to preventive and curative treatment with antagonist bacteria, *Pseudomonas chlororaphis* PCL1606 (green) and fermented *Bacillus velezensis* UMAF6639 (red), and sterile water with adjuvant was used as a control (blue). Statistical analysis was conducted by Student's *t*-test using SPSS software, and significant differences are marked with asterisks; Table S1: Information obtained from fungal isolated ITS and beta-tubulin sequence comparison in NCBI database and was considered for fungal isolated identification.

**Author Contributions:** L.G.-M.: molecular analysis and support in field trials; S.T.: fruit trial support, data curation and statistical analysis; J.A.G.-B. and A.d.V.: conceptualization and supervision; F.M.C.: conceptualization, supervision, and review; E.A.: experimental design and performance, data curation, formal analysis, writing, review and editing. All authors have read and agreed to the published version of the manuscript.

**Funding:** Aid for Knowledge Transfer Activities between Agents of the Andalusian Knowledge System and the Productive Fabric. Junta of Andalusia: AT17_5544_UMA. Ecological Transition and Digital Transition Projects. Ministry of Science and innovation (Spanish Government), European Funds-Recovery, Transformation and Resilience Plan: TED2021-129369B-I00.

**Data Availability Statement:** Data are contained within the article and supplementary materials.

**Acknowledgments:** Special thanks to Alejandro Pérez-García for facilitating the inclusion of *Bacillus* in the study, and to Koppert Biological Systems for supplying the fermented *Bacillus velezensis*-based product UMAF6639 for the study.

**Conflicts of Interest:** The authors declare that they have no known competing financial interests or personal relationships that could have appeared to influence the work reported in this paper.

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
