# Peer review of "Biological Control and Cross Infections of the Neofusicoccum spp. Causing Mango Postharvest Rots in Spain"

_horticulturae, doi:10.3390/horticulturae10020166_

Round 1

Reviewer 1 Report

Comments and Suggestions for Authors

This manuscript deals with an important problem - disease of mango and avocado fruits in Spain. Numerous extensive experiments were carried out, which result with identification of common causal agents of mango rots, it was examined whether these fungi can also occur in avocado fruits and whether cross infection of mango and avocado may occur. The next experiments concerned the possibility of using antagonistic bacteria (Pseudomonas chlororaphis and Bacillus velezensis) to prevent these diseases under controlled and field conditions (preventive and curative variants). The results bring new elements to science and horticultural practice. Therefore this manuscript should be published in Horticulturae. However, this extensive manuscript requires some additions, explanations and corrections according to the examples indicated in Remarks.

Remarks

Title - requires change. You received a positive result for two species of Neofusicoccum, so there is no need to extend it to the level of the Botryosphaeriaceae family. Moreover, not all Botryosphaeriaceae species isolated from mango were tested for their pathogenicity.

Line 16 artificial infection - rather artificial inoculation

Line 32 Ha - rather ha

Line 48-57 reconstruct the text because there are unnecessary repetitions

Line 103 PDA – specify manufacturing company

Line 115 delete ‘biocontrol’ once

Table 1 contains a list of isolates and not 'Characteristics of fungal isolates'.

Table 1 the correct spelling is 'sp.' not italic

Table 1 it should be Xylaria instead of Xylavia

Line 141 In my opinion, it is necessary to distinguish between 'inoculation' and 'infection' throughout the manuscript

Line 146 how do we know at this stage that these were Alternaria spp. and not one Alternaria sp. The same applies to lines 148, 188, 189.

Line 161 you write ‘incubated at room temperature for 7 days’ ; in line 169 and 171 you write eight days - it requires correction

Line 199 can add ‘in dual cultures in vitro’

Line 202 rather ‘dual cultures technique’

Line 203 Provide the formula according to which the degree of fungal diameter reduction under the influence of bacteria in dual cultures was calculated. Moreover, in all experiments in which there were repetitions, it should be provided how the final result was determined

Line 275 pseudomonas, Pseudomonas and on other places make corrections

Line 349 collection of 75 isolates was analyzed in this study, but in line 346 you write that 10 isolates from 2013 were included, so the total should be 85 (or clarify the text)

Line 365 Alternaria spp. – spp. should be not italic

Line 381 no data – how should it be interpreted exactly?

Line 382 Genus Neofusicoccum sp. – in this case sp. must be removed

Line 400 specie??

Line 409 in M&M it is necessary to determine what the criterion was for selecting isolates for this experiment

Figure 2, Table 2 – why are isolate numbers used and not species names? This makes them difficult to read and it takes a lot of time to find what species is hidden under a given number. Adding, for example, Np (for Neofusicoccum parvum) before isolate number would make the information easier to understand

Figure 2 Percentage of growth of isolates - % should be written in the figure 0-100%

Line 423-424 Since you write 'genera' - please delete all spp. in these two lines

Line 438 antagonis'bacteria ???

Line 407 the title of chapter 3.3 should be changed, in its current form it is not known what it is about

Line 414 ‘showed less antagonism to Pc presence’ - what does this mean? The obtained result should be precisely determined

Figure 3 Explanation is not clear, it suggests that Pc and Bv caused rot symptoms on fruit

Line 485-486 the current text is unclear, it should be Neofusicoccum parvum (UMAF M1937 and UMAF M1302) and N. mediterraneum (UMAF M1938).

Figure 4 Why does the result regarding Bacillus amyloliquefaciens appear here, the methodology did not mention this species of bacteria at all

Line 520 it should be … ‘counts (Figure 5).

Line 568 Neofusicoccum spp. – list specific species.

Line 597 it should be B. amyloliquefaciens

Line 609-610 consider revising this sentence

Line 627 ‘infective points – infection points ??

Line 672, 675 spp. – not in italic

Literature – Latin names of plants and fungi should be written in italic

Table S1 it should be Xylaria sp. instead of Xylavia sp.

Comments on the Quality of English Language

see Remarks

Author Response

Responses to Request of Reviewer 1

horticulturae-2844778

“Biological control and cross infections of the Botryosphaeriaceae family of fungi causing mango postharvest rots in Spain.” Lucía Guirado-Manzano, Sandra Tienda, José Antonio Gutiérrez-Barranquero, Antonio de Vicente, Francisco M. Cazorla, Eva Arrebola

Thank you for the revision and remarks made in the manuscript such as the time invested in the article.

Title - requires change. You received a positive result for two species of Neofusicoccum, so there is no need to extend it to the level of the Botryosphaeriaceae family. Moreover, not all Botryosphaeriaceae species isolated from mango were tested for their pathogenicity.

- Thank you for the comment, the title has been changed to “Biological control and cross infections of the Neofusicoccum spp. causing mango postharvest rots in Spain”

Line 16 artificial infection - rather artificial inoculation

- Line 16 has been corrected to “artificial inoculation”. Line 16

Line 32 Ha - rather ha

- Line 32 has been corrected to “ha”. Line 32

Line 48-57 reconstruct the text because there are unnecessary repetitions

- Lines 48-57 have been reconstructed erasing repetitions. Lines 48-56

Line 103 PDA – specify manufacturing company

- Line 103 The specific manufacturing company has been included (line 101)

Line 115 delete ‘biocontrol’ once

- Line 115 the sentence has changed to “For experiments with biocontrol bacteria”, (line 117)

Table 1 contains a list of isolates and not 'Characteristics of fungal isolates'.

Table 1 the correct spelling is 'sp.' not italic

Table 1 it should be Xylaria instead of Xylavia

- Table 1 has been corrected to “List of fungal isolates obtained from mango and avocado skin. Strains codes and ITS accession numbers of the isolates used in this study”. The “sp.” Has been changed to normal text and “Xylavia” has been corrected to “Xylaria

Line 141 In my opinion, it is necessary to distinguish between 'inoculation' and 'infection' throughout the manuscript

- Line 141 A general check of “Inoculation vs infection” has been done according to inoculation is related to the action of putting in contact with the host and pathogen, and infection is related to the development of the fungal growth. The nexts lines have changed: lines 152,165,175,600

Line 146 how do we know at this stage that these were Alternaria spp. and not one Alternaria sp. The same applies to lines 148, 188, 189.

- Line 146, 148, 188 and 189. We use Alternaria spp. because we have fungal isolated with similarities with Alternaria tenuissima, A. alternata, A. pobletensis, A. brassicicola and A. carthami, mostly A. alternata and A. tenuissima but we only have ITS sequence, so we cannot determinate exactly the species.

Line 161 you write ‘incubated at room temperature for 7 days’ ; in line 169 and 171 you write eight days - it requires correction

- Line 161, 169, 171, the text has been corrected to seven days (lines 171 and 174)

Line 199 can add ‘in dual cultures in vitro’

- Line 199, “dual cultures in vitro” has been added (line 201)

Line 202 rather ‘dual cultures technique’

- Line 202 “dual cultures technique” has been changed to “in dual cultures in vitro” (Line 204)

Line 203 Provide the formula according to which the degree of fungal diameter reduction under the influence of bacteria in dual cultures was calculated. Moreover, in all experiments in which there were repetitions, it should be provided how the final result was determined

- Line 203 The formula of the percentage calculation has been added in line 208-212 and the final results used in the figures are detailed in line 346-347 in the Statistical Analysis section.

Line 275, Pseudomonas and on other places make corrections

- Line 275, pseudomonas has been corrected to Pseudomonas. (Lines 238 and 284)

Line 349 collection of 75 isolates was analyzed in this study, but in line 346 you write that 10 isolates from 2013 were included, so the total should be 85 (or clarify the text)

- Line 349, the collection is composed of 34 isolates from mango in 2019 + 10 isolates from mango in 2013 + 31 isolates from avocado in 2019, the total fungi collection is 75 isolates. The information was not easily expressed in the text, so the collection composition has been clarified. (Lines 354-357)

Line 365 Alternaria spp. – spp. should be not italic

- Line 365, The italic has been changed to normal text. (Line 375)

Line 381 no data – how should it be interpreted exactly?

- Line 381. Because in Spain, the mango harvest is done before the avocado harvest, the experiments in mango were done first during the season, and later the avocado experiment was performed. So, there are fungal isolates that were tested only in mango, others only in avocado, and others in both fruits. Thus, there are fungal isolates that were not tested in mango or in avocado, so we don’t have data of that interaction. For that, we classify as “no data” (nd) in the table because it was not assayed (line 391).

Line 382 Genus Neofusicoccum sp. – in this case sp. must be removed

- Line 382, sp. Has been removed (Line 398-399)

Line 400 specie??

- Line 400, Thank for the comment, I mean “Species”, the mistake has been corrected, (line 410)

Line 409 in M&M it is necessary to determine what the criterion was for selecting isolates for this experiment

- Line 409, a paragraph related to the criterion of selected isolates has been added in material and methods, (lines 114-117 and 420-422).

Figure 2, Table 2 – why are isolate numbers used and not species names? This makes them difficult to read and it takes a lot of time to find what species is hidden under a given number. Adding, for example, Np (for Neofusicoccum parvum) before isolate number would make the information easier to understand

- In Figure 2 (444-445) and Table 2, the Np (Neofusicoccum parvum) or Nm (Neofusicoccum mediterraneum) have been added in those isolates whose species has been determined.

Figure 2 Percentage of growth of isolates - % should be written in the figure 0-100%

- Figure 2. The spider chart shows the percentage axis, from 0 (center point) to 100% on the outermost line. The scale is marked in gray. Each spider line corresponds to a percentage range and the values of each fungal growth percentage are placed depending on their value, within one range or another, just as in a graph with two axes. That point has been clarified in the figure legend.

Line 423-424 Since you write 'genera' - please delete all spp. in these two lines

- Lines 423-424, the “spp.” has been removed (lines 436-339)

Line 438 antagonis'bacteria ???

- Line 438, Thanks for the correction, the orthographic mistake has been corrected to “antagonist bacteria” (line 451)

Line 407 the title of chapter 3.3 should be changed, in its current form it is not known what it is about

- Line 407, the title of 3.3 section has been corrected to " Antagonism assays of biocontrol agents Bacillus velezensis and Pseudomonas chlororaphis against fungal isolates.” Also, slight modifications have been made in the following text, which is remarked. (Lines 417-418)

Line 414 ‘showed less antagonism to Pc presence’ - what does this mean? The obtained result should be precisely determined

- Line 414, sorry for the expression, it has been changed to “more tolerance” and the percentage of inhibition has been also included. (Line 426)

Figure 3 Explanation is not clear, it suggests that Pc and Bv caused rot symptoms on fruit

- Figure 3 legend has been corrected for better understanding (Lines 466-468)

Line 485-486 the current text is unclear, it should be Neofusicoccum parvum (UMAF M1937 and UMAF M1302) and N. mediterraneum (UMAF M1938).

- Line 485-486, the text has been corrected to Neofusicoccum parvum (UMAF M1302 and UMAF M1937) and N. mediterraneum (UMAF M1938). (Line 498-499)

Figure 4 Why does the result regarding Bacillus amyloliquefaciens appear here, the methodology did not mention this species of bacteria at all

- Figure 4. Bacillus amyloliquefaciens is the previous nomenclature of Bacillus velezensis, it has been corrected to Bacillus velezensis in Figure 4.

Line 520 it should be … ‘counts (Figure 5).

- Line 520, has been corrected to “bacterial population’ counts (Figure 5)” (Line533)

Line 568 Neofusicoccum spp. – list specific species.

- Line 568, has been to specific species of Neofusicoccum (Line 581)

Line 597 it should be B. amyloliquefaciens

- Line 597, the orthographic mistake has been corrected (Line 611)

Line 609-610 consider revising this sentence

- Line 609-610, the sentence has been corrected (Lines 618-620)

Line 627 ‘infective points – infection points??

- Line 627 the orthographic mistake has been corrected (Line 642)

Line 672, 675 spp. – not in italic

- Line 672-675 spp. has been corrected to normal text (Lines 687, 689-690)

Literature – Latin names of plants and fungi should be written in italic

- Literature, Latin names of plants and fungi have been corrected

Table S1 it should be Xylaria sp. instead of Xylavia sp.

- Table S1, Xylavia has been corrected to Xylaria

Reviewer 2 Report

Comments and Suggestions for Authors

The function of Pseudomonas chlororaphis and Bacillus velezensis used as biocontrol bacteria to prevent fungal infection on mango and avocado fruit was investigate in the manuscript. The topic is interesting and the results provide us a new perspective to develop new biological antagonists. However, the introduction is not sufficient to conclude the background of the study and the discussion did not shed light on the results obtained. Many useful results weren’t analyzed Moreover, the design of experiment is defective and the methods used in the experiment was incorrect. Therefore, I am regretted for having to reject this paper.

Specific comments

1. The title needs to be reconcluded.

2. The abstract should conclude the main results of the study.

3. The introduction should focus on the postharvest disease of mango and avocado caused by  Botryosphaeriaceae family of fungi. In addition, biological control by using microorganisms to prevent postharvest disease of fruit needs to be concluded, sepcially, Pseudomonas chlororaphis PCL1606 and Bacillus velezensis UMAF6639 strain.  

4. Do Alternaria spp. and Neofusicoccum spp. belong to Botryosphaeriaceae family ?

5. Line101-103: ‘From disinfected fruits, small fragments were trimmed, including the border where there was rot, and deposited in acidified potato-dextrose-agar (PDA) plates (PDA + 1 ml of lactic acid), to avoid bacterial development’. The isolation method of fungi from the decayed mango and avocado fruit is unclear. The natural decayed fruit were collected and the decay tissues were taken for the fungi isolation. The whole area of the decay tissues should be taken and mixed with sterilized water, after vortexed and centrifuged, the supernatant should be collected and spread on the PDA plates. After incubation, the mycillium of single colony should be transferred to the new plates for the further isolation. In addition, does lactic acid inhibit all the growth of bacterium ? The detail of the isolation process should be given.

6. It’s better to use graph to exhibit the category of fungi isolated from the fruit.

7. Line 146-147: Why did author choose Alternaria spp. and Neofusicoccum spp. for the fruit inoculation ?

8. In the section ‘2.3 Reproduction of rot symptoms in mango fruit’, ‘hyphal suspensions’ was used for the fruit inoculation. It’s incorrect. Spore suspension is used for fungal inocualtion. In this case, all the fungal inoculation experiment must be performed again.

9. For the Fig. 1, it’s better to make one colonization site on one fruit, not 6 sites. And the fruit without bacterial treatment and inoculated with fungus should be used as the control.

10. In votro experiment should be performed to elucidate the ability of the bacteria preventing fungal growth.  

Comments on the Quality of English Language

Minor editing of English language required

Author Response

Responses to Request of Reviewer 2

horticulturae-2844778

“Biological control and cross infections of the Botryosphaeriaceae family of fungi causing mango postharvest rots in Spain.” Lucía Guirado-Manzano, Sandra Tienda, José Antonio Gutiérrez-Barranquero, Antonio de Vicente, Francisco M. Cazorla, Eva Arrebola

Thank you for the revision and remarks made in the manuscript such as the time invested in the article.

  1. The title needs to be reconcluded
  2. Thank you for your suggestion, the title has been changed to “Biological control and cross infections of the Neofusicoccum sp. causing mango postharvest rots in Spain”.
  3. The abstract should conclude the main results of the study.
  4. The specific species that reproduce rot symptoms in mangoes have been added to the abstract, completing the conclusion of the study. All the conclusions obtained from the main results are now exposed in the abstract (Lines 16-17 and 20-24).
  5. The introduction should focus on the postharvest disease of mango and avocado caused by  Botryosphaeriaceaefamily of fungi. In addition, biological control by using microorganisms to prevent postharvest disease of fruit needs to be concluded, sepcially, Pseudomonas chlororaphis PCL1606 and Bacillus velezensis UMAF6639 strain.
  6. Thank you for your comment about the introduction. This work indeed has two key points, first the identification of the causal agent of rot in mango fruits and another the first approach to control through biological methods. To situate the reader in the current scenario postharvest problems, the introduction section approaches the mango situation in Spain, the postharvest problems, chemical and biological control strategies, and presents two potential biocontrol agents against mango postharvest diseases.
  7. DoAlternaria spp. and Neofusicoccum spp. belong to Botryosphaeriaceae family ?
  8. Certainly Alternatia and Neofusicoccum belong to different Families, Alternaria belongs to the Pleosporaceae Family and Neofusicoccum belongs to the Botryosphaeriaceae Family.

We kept the Alternaria isolates during the first part of our study because it was very frequently isolated from mango fruits. However, Alternaria, even causing some rot, did not reproduce the observed symptoms. From these results, our study focused on pathogenic Neofusicoccum. (Lines 581-582, 666-667).

  1. Line101-103: ‘From disinfected fruits, small fragments were trimmed, including the border where there was rot, and deposited in acidified potato-dextrose-agar (PDA) plates (PDA + 1 ml of lactic acid), to avoid bacterial development’. The isolation method of fungi from the decayed mango and avocado fruit is unclear. The natural decayed fruit were collected and the decay tissues were taken for the fungi isolation. The whole area of the decay tissues should be taken and mixed with sterilized water, after vortexed and centrifuged, the supernatant should be collected and spread on the PDA plates. After incubation, the mycillium of single colony should be transferred to the new plates for the further isolation. In addition, does lactic acid inhibit all the growth of bacterium? The detail of the isolation process should be given.
  2. Thank you for your comment. We have clarified the methodology followed in section 2.1. We hope that the fungal isolation process is now better explained. Lines 99-102

Regarding acidified PDA, it is currently used to avoid most bacterial growth during fungal isolation from environmental samples.

  1. It’s better to use the graph to exhibit the category of fungi isolated from the fruit.
  2. Thank you for the suggestions. We have included a graph exhibiting the categories of fungi isolated from fruit. Supplemented material Figure S2. Lines 117, 677-678.
  3. Line 146-147: Why did author choose Alternariaspp. and Neofusicoccum spp. for the fruit inoculation?

7.- Throughout the sample collection we observed that the presence of Alternaria was practically constant, followed by the isolation of Neofusicoccum, these two genera being the most frequently detected. Both Alternaria and Neofusicoccum are described as pathogens capable of producing diseases in fruits, therefore representative isolates of both genera were tested in the pathogenicity test.

  1. In the section ‘2.3 Reproduction of rot symptoms in mango fruit’, ‘hyphal suspensions’ was used for the fruit inoculation. It’s incorrect. Spore suspension is used for fungal inocualtion. In this case, all the fungal inoculation experiment must be performed again.
  2. Thank you for the comment and we agree that the most correct way to perform an inoculation is using a spore suspension whose concentration can be adjusted. However, the sporulation capacity of Neofusicoccum spp. in a culture medium was very limited. The Neofusicoccum isolates obtained from the samples analyzed during this study do not sporulate in a culture medium. Alternatively, we found several works where a suspension of hyphae was used as inoculum, which are already cited in the manuscript [26,27] line 152, which correspond to:
  3. Oikawa, A.; Ishiara, A.; Tanaka, C.; Mori, N.; Tsuda, M.; Iwamura, H. Accumulation of HDMBOA-Glc is induced by biotic stresses prior to the release of MBOA in maize leaves. Phytochem. 2004, 65:2995-3001. doi: 10.1016/j.phytochem.2004.09.006.

  1. Ahmed, A.O.A.; van de Sande, W.W.J.; van Vianen, W.; van Belkum, A.; Fahal, A.H.; Verbrugh, H.A.; Bakker-Woudenberg, A.J.M. In vitro susceptibilities of Madurella mycetomatis to Itraconazole and Amphotericin B assessd by a modified NCCLS methods and a viability-based 2,3-bis(2-methoxy-4-nitro-5-sulfophenyl)-5-[(Phenylamino)carbonyl]-2H-Tetrazolium hydroxide (XTT) assay. Antimicrob. Agents Ch. 2004, 48(7):2742-2746. doi: 10;1128/AAC.48.7.2746.2004.

One of the publications, (Oikawa et al. 2004) complements the creation of the inoculum as a hyphal suspension with the calculation of colony-forming units of the inoculum, as a way of showing and adjusting its concentration. We have done the same, to report the concentration of the fragments that form or initiate the growth of the fungus (line 157).

  1. For the Fig. 1, it’s better to make one colonization site on one fruit, not 6 sites. And the fruit without bacterial treatment and inoculated with fungus should be used as the control.
  2. Certainly one fruit with one inoculation point would have been ideal. Unfortunately, the quantity and availability of mango fruits were limited. For this, we used in this study 75 mangoes per trial, in total of 225 mangoes. Considering that the mango fruits we were using have a considerable size and we monitored the development of the infection, we considered including several inoculation points for each mango fruit (inoculation-points separation was at least 5 cm), as it has been previously reported to study other postharvest diseases (citation has been included into the manuscript, line 240)

According to the referee's suggestion that fruit without bacterial treatment and inoculated with fungus should be used as control, we completely agree it is one of the controls used in the study (lines 265-272). Since the biocontrol trials were composed of three elements: fruit, bacteria, and fungus; we used three controls to include all the combinations:

  • The mangoes were inoculated with fungi but not treated with bacteria, the application only had water (to simulate the washing effect that the application makes) lines 266-267. This is the control suggested by the referee.
  • Mangoes without fungal inoculation (but we made the wounds and simulated the inoculation using water) but treated with bacteria (lines 267-270)
  • Mangoes where we made the wounds simulating the inoculation using water and treated without bacteria, instead we used water to simulate the washing effect. This was our negative control (Lines 270-272).
  1. In votroexperiment should be performed to elucidate the ability of the bacteria preventing fungal growth. 
  2. We completely agree with the reviewer, it is essential to test the control abilities of the antagonists in vitro before applying them to the fruits or the crop. For that, the inhibition ability of Bacillus velezensis UMAF6639 and Pseudomonas chlororaphis PCL1606 were first tested in vitro using dual cultures in plate (section 2.5, lines 202-232). Additionally, antagonistic compounds produced by the bacteria were analyzed: diffusible (lines 203-213), and volatile organic compounds (VOCs) (lines 219-231), to approach the beneficial bacteria mode of action.

Reviewer 3 Report

Comments and Suggestions for Authors

The manuscript entitled by Guirado-Manzano et al demonstrated a strategy to control postharvest rot induced by fungi in mango and avocado using two types of bacteria, Pseudomonas chlororaphis PCL1606 and Bacillus velezensis UMAF6639. This topic is interesting and quite important for postharvest protection of fruits, especially for those need to be stored for a long period. The results presented could largely support the conclusion, and the manuscript read very well. But, some issues need to be addressed by the authors.

In 3.2, did the author get any conclusion that which isolate(s) is(are) the causal agent of mango and avocado fruit rot disease, especially during the postharvest period? Please clarify this clearly. This is quite important, because those causal agents will be logically used in he fungal antagonism assays.

In 3.3, the author selected 34 representative fungal isolates for the fungal antagonism assay, please clarify how these 34 isolates were picked out?

In the discussion section, I suggest to discuss the potential effect on food safety induced by the two biocontrol agents Bc and Pc. Because these agents are used for disease control on fruits, I thus believe that this is quite interested and important for the readers. 

Author Response

Responses to Request of Reviewer 3

horticulturae-2844778

“Biological control and cross infections of the Botryosphaeriaceae family of fungi causing mango postharvest rots in Spain.” Lucía Guirado-Manzano, Sandra Tienda, José Antonio Gutiérrez-Barranquero, Antonio de Vicente, Francisco M. Cazorla, Eva Arrebola

Thank you for the revision and remarks made in the manuscript such as the time invested in the article.

In 3.2, did the author get any conclusion that which isolate(s) is(are) the causal agent of mango and avocado fruit rot disease, especially during the postharvest period? Please clarify this clearly. This is quite important, because those causal agents will be logically used in the fungal antagonism assays.

In 3.2, Thank you for the comment, according to the results shown in section 3.2 and the isolation frequency, we concluded that Neofusicoccum parvum and N. mediterraneum are the causing agents of stem-end rot in mango during postharvest storage. The list of specific species has been included in the discussion section 4.1 (Lines 582-583)

In 3.3, the author selected 34 representative fungal isolates for the fungal antagonism assay, please clarify how these 34 isolates were picked out?

In 3.3, Thank you for the comment. Due to the number of isolates and the limitation of resources, we decided to choose a smaller group of isolates for the antagonism experiments. To do this, we chose representatives of all the genera of fungi found, and of the most abundant ones, we chose several isolates obtained in different sampling, covering different batches of fruits and times of processing of the samples. This criterion has been added in the Material and Methods section 2.1 (lines 114-116)

In the discussion section, I suggest to discuss the potential effect on food safety induced by the two biocontrol agents Bc and Pc. Because these agents are used for disease control on fruits, I thus believe that this is quite interested and important for the readers

In the discussion section, certainly nothing is said about the food safety of the two antagonists used in the study and it is something that should have been addressed. For this reason, two citations have been added where it is concluded that both Bacillus velezensis (Na et al 2022, cited with number 46) and Pseudomonas chlororaphis (Anderson et al. 2018, cited with the number 47) do not pose any danger to human health. Lines 612-613.